# SCHEDULERS FOR SCHEDULE-FREE: THEORETICALLY INSPIRED HYPERPARAMETERS

## ABSTRACT

The recently proposed `schedule-free` method has been shown to achieve strong performance when hyperparameter tuning is limited. The current theory for `schedule-free` only supports a constant learning rate, where-as the implementation used in practice uses a warm-up schedule. We show how to extend the *last-iterate* convergence theory of `schedule-free` to allow for any scheduler, and how the averaging parameter has to be updated as a function of the learning rate. We then perform experiments showing how our convergence theory has some predictive power with regards to practical executions on deep neural networks, despite that this theory relies on assuming convexity. When applied to the warmup-stable-decay (`wsd`) schedule, our theory shows the optimal convergence rate of $\mathcal{O}(1/\sqrt{T})$. We then use convexity to design a new adaptive Polyak learning rate schedule for `schedule-free`. We prove an optimal *anytime* last-iterate convergence for our new Polyak schedule, and show that it performs well compared to a number of baselines on a black-box model distillation task.

## 1 INTRODUCTION

The recently introduced *schedule-free* method (Defazio et al., 2024) achieves state-of-the-art performance over a range of deep learning problems, as proven by its winning entry for the MLCommons 2024 AlgoPerf Algorithmic Efficiency Challenge Self-Tuning track[1].

The efficacy of `schedule-free` on these highly non-convex deep learning problems is remarkable considered that it was designed for convex losses. Indeed, `schedule-free` achieves the optimal $\mathcal{O}(DG/\sqrt{T})$ convergence rate on the class of convex $G$–Lipschitz losses, for $D := \|\boldsymbol{x}_0 - \boldsymbol{x}_\star\|$, where $\boldsymbol{x}_0$ and $\boldsymbol{x}_\star$ are the first and optimal parameters, respectively.

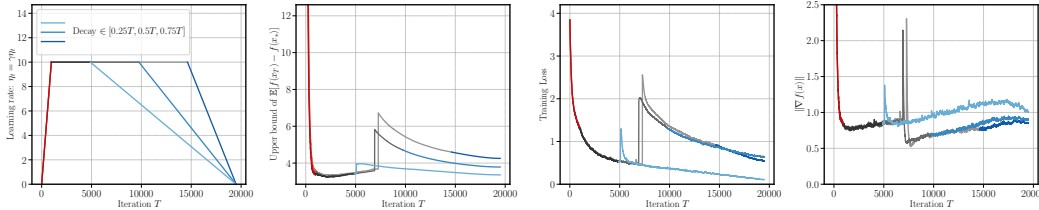

Figure 1: Our theory (Theorem 2.1) is good at predicting the behavior of the training loss: The plots show the theoretical bound and the training loss of `ResNet-20/Cifar10` when using `wsd` schedules with base learning rate $\gamma = 10$ and three different cooldown lengths. The gradient norm over the iteration is shown on the rightmost figure for reference. The red color denotes the warmup period, the gray color denotes the constant period, and the blue color denotes the cooldown period.

We first extend the theory of `schedule-free` to allow for any learning rate scheduler. This is important because the current theory for `schedule-free` in (Defazio et al., 2024) only supports constant learning schedules, where-as in practice `schedule-free` method is applied with a warmup schedule. Although Defazio et al. (2024) has a bound that holds for arbitrary schedules (see Theorem

---

[1] https://mlcommons.org/2024/08/mlc-algoperf-benchmark-competition/

2 in (Defazio et al., 2024)), this bound does not prove convergence. To transform this bound into a convergence theorem, an additional constraint that ties together the learning rates and averaging parameters is required, as we show in Theorem 3.2. When using this new setting for averaging parameters, we refer to the resulting method as `schedulet`. We then specialize our theory to the `wsd` (warmup-stable-decay) schedule and show that `schedulet` achieves the optimal convergence rate of $\mathcal{O}(DG/\sqrt{T})$. We then confirm that our resulting convergence theorem, despite having been established for convex losses, is remarkably good at predicting the behavior of `schedulet` on deep learning tasks. See Figure 1 for a comparison between our theoretical prediction of the loss curve and the empircal loss curve for training a `ResNet-20`.

Second, we propose a new adaptive learning rate for `schedule-free` based on the Polyak step-size, which we call `schedulep`. We establish the last-iterate convergence of `schedulep`, which achieves an *any-time* (meaning that the total number of iterations is not known in advance) optimal convergence rate of $\mathcal{O}(GD/\sqrt{t})$ for every $t$ for the convex and $G$-Lipschitz setting. The downside to `schedulep` is that it requires access to the batch loss on the optimal parameters. Fortunately this optimal loss can be reasonably approximated in either the interpolation setting, or the black-box model distillation setting, in which the student (a smaller model) is trained on one of the tasks that the teacher (a larger model) is pretrained. Under this setting, we can obtain an approximation of optimal batch loss of the student by querying the teacher's loss.

## 1.1 SCHEDULE-FREE SGD

Consider the stochastic optimization problem

$$\min_{\boldsymbol{x} \in \mathbb{R}^d} f(\boldsymbol{x}) := \mathbb{E}_{\mathcal{D}}\left[f_{\boldsymbol{\zeta}}(\boldsymbol{x})\right],$$

where $\mathcal{D}$ is some data distribution over $\mathbb{R}^q$, $\boldsymbol{\zeta} \in \mathbb{R}^q$ is sampled data from $\mathcal{D}$, and $f_{\boldsymbol{\zeta}} : \mathbb{R}^d \to \mathbb{R}$ is our loss function. We assume that $f \colon \mathbb{R}^d \to \mathbb{R}$ is convex, $G$–Lipschitz and that the problem is well-posed, in the sense that a minimizer $\boldsymbol{x}_\star \in \operatorname{argmin}_{\boldsymbol{x} \in \mathbb{R}^d} f(\boldsymbol{x})$ exists.

The `schedule-free` has three sets of iterates, the primal averaging iterates $\boldsymbol{y}_t$, the offline averaging iterates $\boldsymbol{x}_t$, and accumulate gradient iterates $\boldsymbol{z}_t$. At iteration $t$ (for $t = 0, 1, \ldots, T - 1$), we draw a batch of data $\boldsymbol{\zeta}_t$ and evaluate the stochastic gradient[2] $\nabla f(\boldsymbol{y}_t, \boldsymbol{\zeta}_t)$ at $\boldsymbol{y}_t$. At each iteration $t$, this stochastic gradient is used in the `schedule-free` update as follows

$$\boldsymbol{y}_t = (1 - \beta_t)\boldsymbol{z}_{t-1} + \beta_t \boldsymbol{x}_t \tag{1}$$
$$\boldsymbol{z}_t = \boldsymbol{z}_{t-1} - \gamma_t \nabla f(\boldsymbol{y}_t, \boldsymbol{\zeta}_t) \tag{2}$$
$$\boldsymbol{x}_{t+1} = (1 - c_{t+1})\boldsymbol{x}_t + c_{t+1}\boldsymbol{z}_t, \tag{3}$$

where $\beta_t \in [0, 1]$ is the *momentum* parameter, $\gamma_t > 0$ is the *learning rate*, and $c_{t+1} \in [0, 1]$ is the *averaging parameter* over $\boldsymbol{x}_t$ and $\boldsymbol{z}_t$. In practice, the method would be implemented with only one additional sequence given by substituting out $\boldsymbol{y}_t$ as follows

$$\boldsymbol{z}_t = \boldsymbol{z}_{t-1} - \gamma_t \nabla f((1 - \beta_t)\boldsymbol{z}_{t-1} + \beta_t \boldsymbol{x}_t, \boldsymbol{\zeta}_t) \tag{4}$$
$$\boldsymbol{x}_{t+1} = (1 - c_{t+1})\boldsymbol{x}_t + c_{t+1}\boldsymbol{z}_t. \tag{5}$$

The momentum parameter $\beta_t$ interpolates between Polyak-Ruppert averaging when $\beta_t = 0$ and Primal averaging when $\beta_t = 1$. Defazio et al. (2024) suggests that the momentum parameter $\beta_t \equiv \beta \approx 0.9$ works best in practice.

## 1.2 CONTRIBUTIONS AND BACKGROUND

**Schedule-free theory.** Defazio et al. (2024) showed `schedule-free` achieves the optimal $\mathcal{O}(DG/\sqrt{T})$ convergence rate for a fixed horizon $T$ in the convex and $G$-Lipschitz setting with a constant learning rate $\gamma_t \equiv \gamma$ and averaging parameters $c_t = 1/t$ for $t = 1, \ldots, T$. Though Defazio et al. (2024) present a more general result in their Theorem 2 that does hold for every $c_t$ and schedule $\gamma_t$, their result does not guarantee convergence.

---

[2]Formally $\nabla f(\boldsymbol{y}_t, \boldsymbol{\zeta}_t)$ is a subgradient, since we assumed $f(\boldsymbol{y}, \boldsymbol{\zeta})$ is convex in $\boldsymbol{y}$, but not necessarily smooth. But for the sake of simplicity we omit this technical detail.

The `schedule-free` method is also closely related to the AC-SA algorithm, which also converges at the optimal rate of $\mathcal{O}(1/\sqrt{T})$ (Lan, 2012, Corollary 1).

Recently, Brown (2025) proved the convergence of `schedule-free` in smooth nonconvex setting. In all of the cases, the author only discussed the momentum parameter being $\beta_t \equiv 1$, which reduces to the primal averaging. Also, the author considered a constant learning rate $\gamma_t \equiv \gamma$ (or an increasing learning rate $\gamma_t = \gamma_0(t + 1)$) with different choices of $c_t$ for $t = 1, \ldots, T$, and established the best-iterate (in hindsight) convergence to a stationary point.

*Contributions.* We provide a convergence theorem for `schedule-free` in the convex Lipschitz setting that admits any learning rate schedule in Theorem 3.2. To establish this theorem, we require setting the averaging parameter $c_t$ based on the learning rate via $c_t = \gamma_t / \sum_{k=1}^{t} \gamma_k$. In the special case that $\gamma_t$ is constant, this recovers the $c_t = 1/t$ from Defazio et al. (2024). Our theory can be applied to the `wsd` schedule, which yields the optimal convergence rate of $\mathcal{O}(DG/\sqrt{T})$, see Corollary 2.3.

**Momentum for Non-smooth Convex Optimization.** Both Tao et al. (2018) and Defazio & Gower (2021) established that `SGD` with momentum achieves the optimal last-iterate $\mathcal{O}(1/\sqrt{T})$ convergence rate in the convex and Lipschitz setting with a constant step size.

*Contributions.* Because primal averaging is a special case of `schedule-free` when $\beta_t \equiv 1$, and primal averaging itself is equivalent to `Momentum` (see Sebbouh et al. (2021)), our Theorem 3.2 and subsequent Corollary 2.3 for `wsd` schedules includes `Momentum` as a special case. Thus we have extended the convergence of `Momentum` from constant schedules to any schedule.

**Convex Theory for Deep Learning.** Surprisingly, convex optimization theory has been shown to produce practical methods for training large language models. For example, `Adagrad` was developed based on non-smooth convex analysis and became widely used in deep learning until `RMSprop` and `Adam` improved upon it (Duchi et al., 2011). Furthermore, a recent work by Schaipp et al. (2025) has shown that non-smooth convex analysis for `SGD` can effectively predict the performance in deep learning. In particular (Schaipp et al., 2025) found that the empirical convergence of `AdamW` with a `wsd` schedule for large language model training behave similarly to an optimal last-iterate convergence bound for `SGD` in non-smooth convex setting (Defazio et al., 2023).

*Contributions.* Taking inspiration from Schaipp et al. (2025), we compare our new last-iterate convergence theory of `schedule-free` to the practical convergence on a `Resnet-20` for `CIFAR10`. Our comparison shows that the theory can predict which schedules will converge, which schedules will produce spikes with remarkable accuracy, and even when divergence will occur.

**Warmup, stable, decay schedule.** The `wsd` schedule consists of three phases: warmup, constant and cooldown, and hence is also known as the trapezoidal schedule (Zhai et al., 2022). The experiments by Hägele et al. (2024) found that `wsd` performed as good as or even better than the `cosine` schedule with the cooldown phase. Furthermore, `wsd` is better suited for training foundation models, where the cooldown phase can be used for finetuning (Hägele et al., 2024).

*Contributions.* As a special case of our main theorem, we show that, the `schedule-free` SGD method, applied with the `wsd` schedule, can achieve an optimal convergence rate of $\mathcal{O}(1/\sqrt{T})$.

**Polyak Stepsize.** The Polyak stepsize was first introduced by Polyak (1987) in the deterministic setting, where the convergence was proved for the non-smooth and convex setting. Hazan & Kakade (2019) revisited the Polyak stepsize for the class of gradient descent methods and showed that Polyak stepsize has near-optimal convergence rate in the Lipschitz, smooth, and strongly convex setting without accessing to any of the Lipschitz, smoothness or strong convexity parameters.

Recently, there have been many proposals of a stochastic Polyak stepsize in machine learning; see (Berrada et al., 2020; Loizou et al., 2021). Assuming access to $f_\zeta(x_\star)$, the `SPS`$\star$ by Gower et al. (2025) achieves the best known rates across several classes of convex functions. Moreover, Gower et al. (2025) proposed an adaptive Polyak stochastic stepsize, called `IAM` (Iterate Averaging Adaptive method), for the momentum method. Other variants of stochastic Polyak with momentum include (Oikonomou & Loizou, 2024; Wang et al., 2023; Orvieto & Xiao, 2024).

*Contributions.* We suggest a Polyak stepsize for `schedule-free`. With an arbitrary choice of the momentum parameter $\beta_t \equiv \beta \in [0, 1)$, we prove an optimal *anytime* last-iterate convergence bound of $\mathcal{O}(GD/\sqrt{t})$ for every $t$ for the non-smooth convex setting in Theorem 3.2. We then consider the application black-box model distillation setting proposed by Gower et al. (2025), and show that our new Polyak stepsize for `schedule-free` achieves strong performance compared to several benchmark methods on both the `TinyShakespeare` and `fineweb1B` data set.

## 2 CONVERGENCE ANALYSIS AND IMPLICATIONS

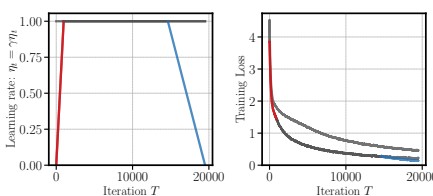

Figure 2: Training loss for `schedule-free` on `ResNet-20` `/Cifar10` with a constant learning rate schedule (gray), warmup-stable (red-gray), and `wsd` schedule (red-gray-blue).

The `schedule-free` algorithm was designed to perform well without the need to tune additional hyper-parameters beyond momentum. For convex and Lipschitz objectives, it achieves the optimal convergence guarantees with a constant step size. Despite this, `schedule-free` is used with a linear warmup schedule, which the authors note is necessary for competitive performance. This added benefit over a constant schedule is demonstrated in Figure 2 for a small deep learning model. This indicates a gap between theory and practice, which motivates a natural question: Does `schedule-free` remain optimal with a non-constant schedule, in the convex setting?

We begin by stating our convergence result for the `schedule-free` SGD method with a general, non-constant learning rate in Theorem 2.1. The proof of the theorem is deferred to Appendix A.

**Theorem 2.1.** Let $f: \mathbb{R}^d \to \mathbb{R}$ be convex and $G$-Lipschitz continuous. Let $\{x_t, y_t, z_t\}$ be generated from (1), (2), (3). Suppose that

$$c_t = \frac{\gamma_t}{\sum_{i=0}^{t} \gamma_i} \tag{6}$$

for $t = 1, \ldots, T$. Initializing $z_{-1} = x_0$, we then have

$$\mathbb{E}\left[f(x_T) - f(x_\star)\right] \leq \frac{\frac{1}{2}\|x_0 - x_\star\|^2 + \gamma_0(f(x_0) - f(x_\star))}{\sum_{t=0}^{T} \gamma_t} + \sum_{t=0}^{T} \frac{\frac{1}{2}\gamma_t^2 G^2}{\sum_{t=0}^{T} \gamma_t}. \tag{7}$$

Our theory shows a last-iterate convergence bound for the `schedule-free` SGD method with general learning rates. First, for $D := \|x_0 - x_\star\|$, we can see that by choosing $\gamma_t \equiv \frac{D}{G\sqrt{T}}$ for all $t$, we recover the optimal $\mathcal{O}(DG/\sqrt{T})$ convergence rate given in Theorem 1 in Defazio et al. (2024). Moreover, for non-constant learning rates, Theorem 2.1 suggests a theoretically well-motivated averaging parameter $c_t$ that is set based on all the past learning rates $\{\gamma_0, \ldots, \gamma_t\}$. This choice of $c_t$ in (6) is similar to the heuristic choice of $c_{t+1} = \frac{\gamma_t^2}{\sum_{i=0}^{t} \gamma_i^2}$ suggested by Defazio et al. (2024, equation (23)). This heuristic choice is the default setting in the code base[3] for `schedule-free`.

A natural question is, why should we care about this theoretical convergence theory which holds only for convex functions, where-as `schedule-free` is a method for non-convex deep learning? Towards answering this question, we perform several experiments comparing the predicted convergence of this theorem, to the practical convergence for training a neural network in the following section.

### 2.1 SURPRISING PREDICTIVE POWER FOR DEEP LEARNING

Inspired by Schaipp et al. (2025), we compute our last-iterate convergence bound from Theorem 2.1 and compare it to the empirical performance of `schedule-free` on `ResNet-20/Cifar10` for `wsd` schedule with cooldown starting at $\{0.25T, 0.5T, 0.75T\}$ where $T$ is the training horizon. We outline the experiment setup and present a comparison using the `cosine` schedule in Appendix E.

---

[3]https://github.com/facebookresearch/schedule_free

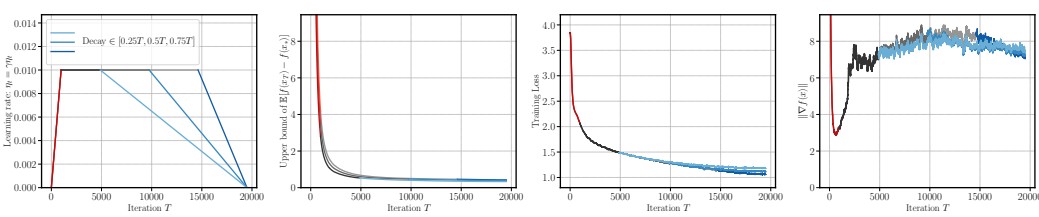

Figure 3: Using `wsd` schedules with three different cooldown periods and with base learning rate $\gamma = 0.01$, our plots compare the theoretical convergence (Theorem 2.1) to the empirical convergence of `ResNet-20/Cifar10`, with the gradient norm shown for reference. The red color denotes the warmup period, the gray color denotes the constant period, and the blue color denotes the cooldown period.

We take $x_\star$ to be the iterate with the smallest loss $f(x_\star)$ during training. In Figures 1 and 3, we use the `wsd` schedule with a large ($\gamma = 10$) and small ($\gamma = 0.01$) base learning rate, respectively.

For a small base learning rate, the theory predicts the convergence seen in practice across all three cooldown schedules, see Figure 3. For a large base learning rate, the theory predicts the transient spikes in the loss regardless whether it occurs *before* or *after* the cooldown period, see Figure 1. One possible explanation is that, the spikes are caused by the spikes in the gradient norm (see the rightmost figure in Figure 1). Yet, one should also note that our theory predicts the convergence in Figure 3 even with the blowup of the gradient norms. Finally, in Figure 4, using a constant-then-diverging schedule, our theory also predicts all spikes in the loss, and whether and when the training diverges.

These experiments show a striking similarity between the convex theory bounds and the loss curves observed in the non-convex setting. Having established that our theory has some predictive power for deep learning , we now specialize our theory to the `wsd` schedule.

## 2.2 APPLICATION TO WSD SCHEDULE

The `wsd` schedule (warmup-stable-decay), a trapezoidal shape learning rate schedule, has been shown to be very useful for training large language models (Hägele et al., 2024). For this section we divide the learning rate into

$$\gamma_t = \gamma \, \eta_t$$

where $\gamma > 0$ is the *base learning rate*, which is the parameter that is tuned, and $\eta_t$ is the *schedule*. For `wsd` there are three phases of the schedule: first, a warmup period, then a constant period, and at last, a cooldown period. Formally, for $0 \le T_w \le T_c \le T$, the `wsd` schedule is given by:

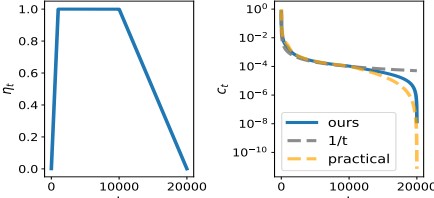

$$\eta_t = \begin{cases} \frac{t+1}{T_w+1}, & \text{if } 0 \le t \le T_w, \\ 1, & \text{if } T_w < t \le T_c, \\ \frac{T-t+1}{T-T_c+1}, & \text{if } T_c < t \le T. \end{cases} \quad (8)$$

Figure 5: The averaging parameter $c_t$ when applied with the `wsd` schedule where blue is our proposed $c_t = \eta_t / \sum_{i=0}^{t} \eta_i$, gray is $c_t = 1/t$, and the orange is the practical heuristic $c_t = \eta_t^2 / \sum_{i=0}^{t} \eta_i^2$.

Substituting the `wsd` schedule (8) into (6), we can obtain a sequence of averaging parameters.

**Lemma 2.2.** Let $0 \le T_w \le T_c \le T$ and $\gamma > 0$. Suppose that $\{\eta_t\}_{t=0}^{T}$ follows the `wsd` schedule given in (8). We can determine $\{c_t\}_{t=0}^{T}$ by

$$c_t = \begin{cases} \frac{2}{t+2}, & \text{if } 0 \le t \le T_w, \\ \frac{2}{2t-T_w+2}, & \text{if } T_w < t \le T_c, \\ \frac{2(T-t+1)}{(T-T_c+1)(2T_c-T_w+2)+(2T-T_c-t+1)(t-T_c)}, & \text{if } T_c < t \le T. \end{cases} \quad (9)$$

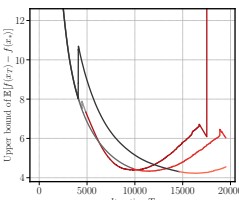 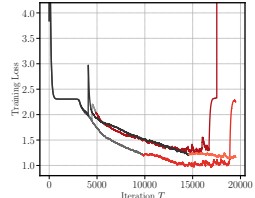 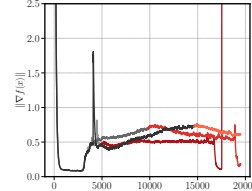

Figure 4: Using schedules with three three different diverging periods, we compare the theoretical convergence given by Theorem 2.1 to the empirical convergence of `ResNet-20/Cifar10`. The gray color denotes the constant period and the red color denotes the diverging period.

To illustrate the results in Lemma 2.2, Figure 5 plots the resulting averaging parameters $c_t$ when applied a `wsd` schedule. The blue line, the gray dashed line, and the orange dashed line depict our proposed $c_t$ in (9), the theoretical $c_t = 1/t$, and the practical default $c_{t+1} = \frac{\gamma_t^2}{\sum_{i=0}^t \gamma_i^2}$ for `schedule-free` (Defazio et al., 2024, Algorithm 1), respectively. As can be seen, our theoretically motivated choice is close to the default practical choice proposed in Defazio et al. (2024), particularly as $t$ grows. Using the `wsd` schedule $\eta_t$ defined in (8) and the weights $c_t$ given in (9), we obtain the convergence result in Corollary 2.3.

Corollary 2.3 shows that, the `schedule-free` SGD with the `wsd` schedule achieves an optimal convergence rate of $\mathcal{O}(1/\sqrt{T})$ as long as the base learning rate is well-chosen.

**Corollary 2.3.** Let $D = \|\boldsymbol{x}_0 - \boldsymbol{x}_\star\|$. Using the `wsd` parameters $(c_t, \eta_t)$ given in (9) and (8), with a base learning rate of $\gamma = \frac{D}{G\sqrt{\sum_{t=0}^T \eta_t^2}}$, we have the convergence

$$\mathbb{E}\left[f(\boldsymbol{x}_T) - \inf f\right] \leq \frac{2\eta_0(f(\boldsymbol{x}_0) - f(\boldsymbol{x}_\star))}{T + T_c - T_w + 2} + \frac{2\sqrt{\frac{2}{3}}DG}{\sqrt{T + T_c - T_w + 2}} \simeq \mathcal{O}\left(\frac{DG}{\sqrt{T}}\right). \quad (10)$$

## 3 POLYAK LEARNING RATE

Having seen that using convexity as an assumption can result in theory with some predictive power on neural network experiments, in this section, we use convexity to design an adaptive learning rate schedule, which we call `schedulep`, see Algorithm 1 for the pseudo-code. Here, we denote $(\cdot)_+^2 = ((\cdot)_+)^2$; i.e., $(a)_+^2 = a^2$ if $a > 0$ and $(a)_+^2 = 0$ otherwise.

To derive this adaptive stepsizes, we make use of the following *Interpolation* assumption.

**Assumption 3.1** (Interpolation). For every $\boldsymbol{\zeta}$, we have access to $f_\zeta(\boldsymbol{x}_\star)$ where $\boldsymbol{x}_\star \in \underset{\boldsymbol{x} \in \mathbb{R}^d}{\operatorname{argmin}} f(\boldsymbol{x})$.

We call this the interpolation assumption, because it holds for models that interpolate the data, in which case $f_\zeta(\boldsymbol{x}_\star) = 0$ since every data point has a perfect fit, and thus zero loss (Ma et al., 2018; Liu et al., 2022; Gower et al., 2021). Many vision models interpolate the data, unlike language models which have a strictly positive entropy rate: the next word in a sequence is never perfectly predictable (Shannon, 1948; Cover & King, 1978). Though one can still approximate $f_\zeta(\boldsymbol{x}_\star)$ for language models, see Section 4.2.

---

**Algorithm 1** Schedulep: Schedule-free Polyak

1: **Input:** $\boldsymbol{z}_{-1} = x_0 \in \mathbb{R}^d, \beta \in [0, 1], c_t > 0, \gamma_{\max} > 0$.
2: **for** $t = 0$ to $T - 1$ **do**
3:     $\boldsymbol{y}_t = (1 - \beta)\boldsymbol{z}_{t-1} + \beta\boldsymbol{x}_t$
4:     $\tau_t = \frac{[f_{\zeta_t}(\boldsymbol{y}_t) - f_{\zeta_t}(\boldsymbol{x}_\star) + \beta\langle\nabla f(\boldsymbol{y}_t, \boldsymbol{\zeta}_t), \boldsymbol{z}_{t-1} - \boldsymbol{x}_t\rangle]_+}{\|\nabla f(\boldsymbol{y}_t, \boldsymbol{\zeta}_t)\|^2}$
5:     $\gamma_t = \min\left\{\gamma_{\max}, \tau_t\right\}$
6:     $\boldsymbol{z}_t = \boldsymbol{z}_{t-1} - \gamma_t \nabla f(\boldsymbol{y}_t, \boldsymbol{\zeta}_t)$
7:     $\boldsymbol{x}_{t+1} = (1 - c_{t+1})\boldsymbol{x}_t + c_{t+1}\boldsymbol{z}_t$
8: **end for**
9: **Return:** $\boldsymbol{x}_T$

We derive our adaptive learning rate by choosing $\gamma_t$ that will bring iterate $z_t$ closer to the solution $x_\star$. For this note from (2) (or equivalently line 1 in Algorithm 1), the iterate $z_t$ explicitly depends on the learning rate $\gamma_t$. Consequently we can write $z_t(\gamma_t) \equiv z_t$. We then derive an upper bound on $\|z_t(\gamma_t) - x_\star\|^2$ that only depends on known quantities and $f_{\zeta_t}(x_\star)$ by assuming that the loss function is convex. Minimizing this upper bound in $\gamma_t$ gives our adaptive learning rate on line 1 in Algorithm 1. We call our resulting algorithm `schedulep` (`schedule-free` with a Polyak learning rate), since this is a generalization of the Polyak learning rate to `schedule-free`. We include this additional cap of $\gamma_{\max}$ on line 1 in Algorithm 1 to improve stability, specially in the case where $f_{\zeta_t}(x_\star)$ is misspecified. This is a common safe-guard used in stochastic Polyak methods (Loizou et al., 2020).

Next we prove the convergence of our `schedulep` method.

**Theorem 3.2.** Consider the iterates of Algorithm 1 with $c_t = 1/(t+1)$, $\beta \in [0,\ 1)$ and $\gamma_{\max} = \infty$. Let $f_\zeta \colon \mathbb{R}^d \to \mathbb{R}$ be a convex function for every $\zeta$. Let

$$B \ := \ \{x \ : \ \|x - x_\star\| \leq \|x_0 - x_\star\|\} \subset \mathbb{R}^d, \tag{11}$$

$$G^2 \ := \ \max_{x \in B} \mathbb{E}_\zeta \|\nabla f(x, \zeta)\|^2. \tag{12}$$

With the initialization $z_{-1} = x_0$, the suboptimality gap of the *last iterate* $x_t$ converges at a $1/\sqrt{t}$ rate according to

$$\mathbb{E}\left[f(x_t) - f(x_\star)\right] \ \leq \ \frac{G\|x_0 - x_\star\|}{\sqrt{t+1}}. \tag{13}$$

The resulting rate of convergence of `schedulep` in (13) is exactly the optimal rate for the class of convex and $G$–Lipschitz functions. Furthermore, this convergence has two additional benefits. First, it is an *anytime* result, in that (13) achieves the optimal rate for every $t$, where-as previous results for `schedule-free` only achieve the optimal $\mathcal{O}(1/\sqrt{T})$ with the known stopping time $T$. Second, we do not need to assume that the loss is globally Lipschitz. Rather, that it is Lipschitz in the closed ball given in (11). Thus we are also able to weaken the global Lipschitz assumption.

## 4 EXPERIMENTS

Our theory suggests a new choice of $c_t$, which we evaluate against the practical heuristic $c_{t+1} = \gamma_t^2 / \sum_{i=1}^t \gamma_i^2$ and that of the previous theory, $c_t = 1/t$. We run experiments from small- to large-scale across domains (vision and language) and learning tasks (regression, classification, and knowledge distillation). For regression and image classification, we use the `SGD` variant of `schedule-free`; for distillation in language modeling, we use the `AdamW-schedulefree` variant in Defazio et al. (2024). We use the momentum parameter $\beta = 0.9$ throughout our experiments.

### 4.1 IMAGE CLASSIFICATION

We test `schedulet` on image classification with `Wide Resnet (16-8)` on `CIFAR10` and `DenseNet` on `CIFAR100`. Hyperparameter settings follow that of Defazio et al. (2024), with exact settings listed in Appendix E and Table 1. We compare the performance of `schedule-free` with `schedulet`, the practical heuristic $c_{t+1} = \gamma_t^2 / \sum_{i=1}^t \gamma_i^2$, $c_t = 1/t$ from previous theory, and `SGD-m` (stochstic gradient descent with momentum). We apply the warmup-stable schedule for `schedule-free` with the practical heuristic averaging parameters and the `wsd` schedule otherwise. We use a 5% warmup for all schedules, and set the cooldown in `wsd` to 25% in smaller models and 5% for larger models. For each model, we sweep the learning rate over a grid for all optimizers, tuning each method using the validation loss as a proxy for generalization ability of the optimizer.

As mentioned in Defazio et al. (2024), `Schedule-free` requires batch statistics computed from the $x$ sequence (i.e. Equation 3) for models using BatchNorm layers. We avoid this complication by using GroupNorm layers for all models, which does not significantly effect the performance and training dynamics of these relatively smaller models.

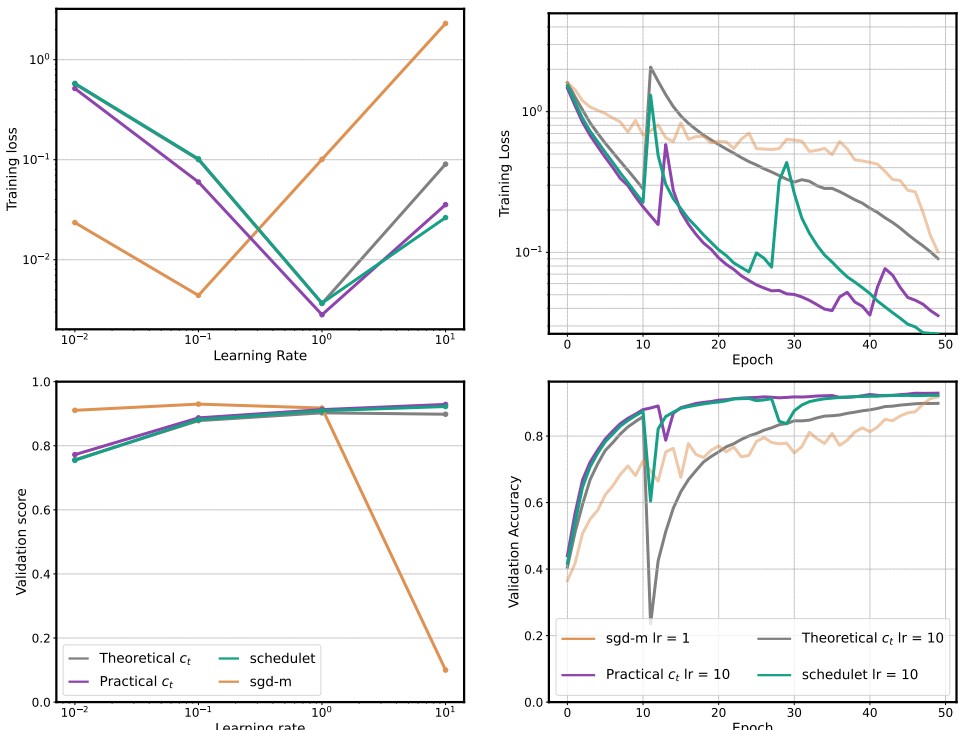

Figure 6: Training a `Wide ResNet (16-8)` model on the `CIFAR10` data set.

The results in Figure 6 show that, although the practical heuristic averaging parameter generally achieves a smaller training loss than `schedulet` across different learning rates, their performance are similar in terms of generalization. Moreover, we see that `schedule-free` with different averaging parameters is robust across different learning rates in terms of validation score. When considering the best tuned learning rate ($\gamma = 1$ for `SGD-m` and $\gamma = 10$ for `schedule-free`), we see that `schedule-free` with the practical heristic $c_t$ performs slightly better than `schedulet` in terms of training loss, but as well with respect to validation score. Yet, they both outperform the choice of $c_t = 1/t$ from previous theory. When training larger models, our experiments show that `schedulet` has similar performance as the practical heuristic parameter; see Figure 10 in Appendix E.1.2.

## 4.2 MODEL DISTILLATION

Here we test `Schedulep` in Algorithm 1 on black-box model distillation, where we have only access to the teacher's loss over a given batch. We will use the teacher's loss as an approximation of the optimal student's loss. That is, let $f_\zeta^t$ and $f_\zeta^s(x)$ denote the teacher's loss and the student's loss with weights $x$, respectively, for a given batch $\zeta$. We will choose a teacher that has been trained on a large corpora, such that $f_\zeta^t \approx f_\zeta^s(x_\star)$, where $x_\star$ are the optimal parameters for the student model.

Our setup is based on the experiments by Gower et al. (2025). As a baseline, we used `SGD-m`, `AdamW` (Kingma & Ba, 2014), `(AdamW-)ScheduleFree` (Defazio et al., 2024), and `IAMS(-Adam)` (Gower et al., 2025). We also test the `AdamW` version of `Schedulep` called `AdamW-Schedulep`, see Appendix C and Algorithm 2 for details. For the distillation experiments, we considered two settings:

**Distilling `tiny_shakespeare`.** The teacher model employed was `gpt2-medium` (345 million parameters), a pre-trained transformer model from the Hugging Face library (Radford et al., 2019). We used a student model with 67.7 million parameters, see Table 2 in Appendix E.2 for details. The results in Figure 7 show that our `AdamW-Schedulep` achieves the best loss for a tuned

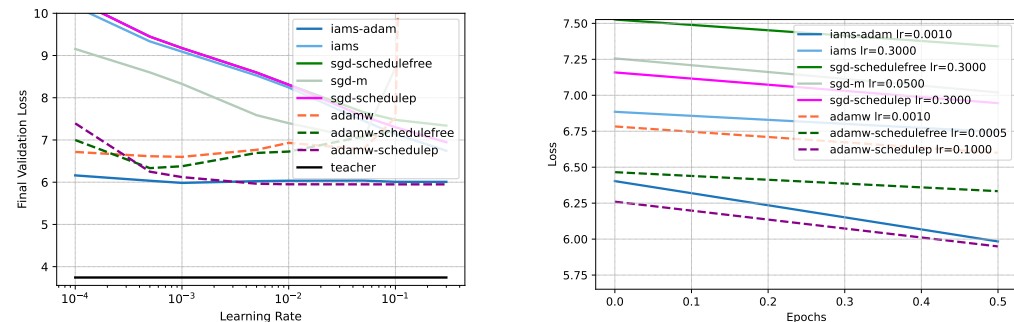

Figure 7: Training a smaller student model on the `tiny_shakespeare` data set, using `gpt2-medium` as the teacher.

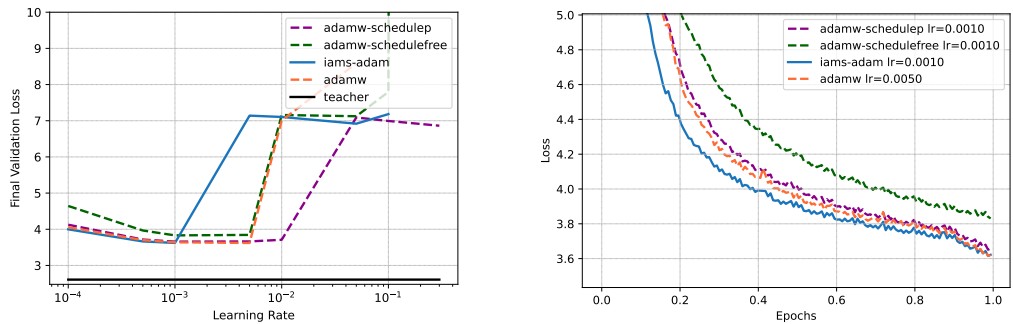

Figure 8: Training a nanoGPT student model on the `fineweb1B` data set, using `EleutherAI/gpt-j-6B` as the teacher.

learning rate $\gamma_{\max}$, but it is not quite as robust as the `IAMS-Adam` method is to the choice of learning rate.

**Distilling `fineweb1B`.** The teacher model employed was `EleutherAI/gpt-j-6B`, a 6-billion parameter transformer model pre-trained on diverse datasets (Wang & Komatsuzaki, 2021). We used a nanoGPT model with 124 million parameters as the student, see Table 2 in Appendix E.2 for details, and Figure 8 shows `AdamW-Schedulep` is now the most robust method with respect to different choices of learning rate $\gamma_{\max}$, but the best loss is achieved by tuning `AdamW` or `IAMS-Adam`.

## 5 CONCLUSION AND LIMITATIONS

We developed the last-iterate convergence theory for `schedule-free` that works for general non-constant schedule in the convex Lipschitz setting. The theory requires the averaging parameter to be a function of the learning rate schedule, which we called `schedulet`. We showed that our theory is good at predicting the empirical behavior of `schedulet`. We also obtained the optimal convergence $\mathcal{O}(GD/\sqrt{T})$ from the theory when specialized to `wsd` schedule. Next, assuming convexity and interpolation, we developed a Polyak stepsize for `schedule-free`, called `schedulep`. We proved an any-time convergence $\mathcal{O}(GD/\sqrt{t})$ for `schedulep` and demonstrated its strong performance compared to several benchmark methods under the black-box distillation model setting.

The limitation of our work is that, our theory only applies for general learning rate schedule with `schedulet`, so it does not give any convergence bounds for the averaging parameter used in practice. In fact, our suggested averaging parameter schedule does not improve the training performance in practice. Moreover, our comparison between the convergence theory and the empirical performance is via visual inspection but not a quantitative analysis. For the Polyak stepsize `schedulep`, it can only be applied to models that nearly interpolate the data or under the black-box model distillation setting.

**Reproducibility Statement.** To ensure reproducibility, we provide our open-source repository built upon publicly available implementations of common vision and language models, optimizers, and training frameworks. We extend the open-source framework `step-back`[4] to incorporate `Schedule-free`[5], `Wide ResNet`[6] and `DenseNet`[7] architectures with GroupNorm layers. Complete training specifications, architectures, and hyperparameters are detailed in Tables 1–2 and Appendix E.

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

CONTENTS

## A   CONVERGENCE THEORY FOR `schedule-free` SGD

### A.1   AUXILIARY LEMMAS

**Lemma A.1.** Let $\{\boldsymbol{x}_t, \boldsymbol{y}_t, \boldsymbol{z}_t\}$ be generated from (1), (2), (3). For $t = 0, 1, \ldots, T-1$, we have the following inequality holds:

$$\frac{1}{c_{t+1}} f(\boldsymbol{x}_{t+1}) - \left(\frac{1}{c_{t+1}} - 1\right) f(\boldsymbol{x}_t) - f(\boldsymbol{x}_\star) \leq \langle \nabla f(\boldsymbol{y}_{t+1}), \boldsymbol{z}_t - \boldsymbol{x}_\star \rangle \tag{14}$$

*Proof.* Dividing both sides of (3) by $c_{t+1}$ and rearranging terms, we have

$$\left(\frac{1}{c_{t+1}} - 1\right)(\boldsymbol{x}_{t+1} - \boldsymbol{x}_t) = \boldsymbol{z}_t - \boldsymbol{x}_{t+1}; \tag{15}$$

and also (1) implies

$$\boldsymbol{z}_t - \boldsymbol{y}_{t+1} = \frac{\beta_{t+1}}{1 - \beta_{t+1}}(\boldsymbol{y}_{t+1} - \boldsymbol{x}_{t+1}), \tag{16}$$

for $t = 0, 1, \ldots, T-1$. Applying (15) and (16) and the fact that $f$ is convex, we then obtain

$$\frac{1}{c_{t+1}} f(\boldsymbol{x}_{t+1}) - \left(\frac{1}{c_{t+1}} - 1\right) f(\boldsymbol{x}_t) - f(\boldsymbol{x}_\star)$$

$$= \left(\frac{1}{c_{t+1}} - 1\right)(f(\boldsymbol{x}_{t+1}) - f(\boldsymbol{x}_t)) + (f(\boldsymbol{x}_{t+1}) - f(\boldsymbol{x}_\star))$$

$$\leq \left(\frac{1}{c_{t+1}} - 1\right)\langle \nabla f(\boldsymbol{x}_{t+1}), \boldsymbol{x}_{t+1} - \boldsymbol{x}_t \rangle + (f(\boldsymbol{x}_{t+1}) - f(\boldsymbol{x}_\star))$$

$$\overset{(15)}{=} \langle \nabla f(\boldsymbol{x}_{t+1}), \boldsymbol{z}_t - \boldsymbol{x}_{t+1} \rangle + (f(\boldsymbol{x}_{t+1}) - f(\boldsymbol{y}_{t+1})) + (f(\boldsymbol{y}_{t+1}) - f(\boldsymbol{x}_\star))$$

$$\leq \langle \nabla f(\boldsymbol{x}_{t+1}), \boldsymbol{z}_t - \boldsymbol{x}_{t+1} \rangle + \langle \nabla f(\boldsymbol{x}_{t+1}), \boldsymbol{x}_{t+1} - \boldsymbol{y}_{t+1} \rangle + \langle \nabla f(\boldsymbol{y}_{t+1}), \boldsymbol{y}_{t+1} - \boldsymbol{x}_\star \rangle$$

$$= \langle \nabla f(\boldsymbol{x}_{t+1}) - \nabla f(\boldsymbol{y}_{t+1}), \boldsymbol{z}_t - \boldsymbol{y}_{t+1} \rangle + \langle \nabla f(\boldsymbol{y}_{t+1}), \boldsymbol{z}_t - \boldsymbol{x}_\star \rangle$$

$$\overset{(16)}{=} \frac{\beta_{t+1}}{1 + \beta_{t+1}} \langle \nabla f(\boldsymbol{x}_{t+1}) - \nabla f(\boldsymbol{y}_{t+1}), \boldsymbol{y}_{t+1} - \boldsymbol{x}_{t+1} \rangle + \langle \nabla f(\boldsymbol{y}_{t+1}), \boldsymbol{z}_t - \boldsymbol{x}_\star \rangle, \tag{17}$$

where the third and the fifth lines have applied the convexity of $f$. Now, because of the convexity of $f$, observe that for any $a, b \in \mathbb{R}^d$,

$$f(a) \geq f(b) + \langle \nabla f(b), a - b \rangle$$
$$f(b) \geq f(a) + \langle \nabla f(a), b - a \rangle.$$

Summing together the two above inequalities gives

$$\langle \nabla f(a) - \nabla f(b), b - a \rangle \leq 0$$

and thus the first term of (17) is negative. This then completes the proof of the lemma. □

**Lemma A.2.** Let $\{\boldsymbol{x}_t, \boldsymbol{y}_t, \boldsymbol{z}_t\}$ be generated from (1), (2), (3). Suppose that

$$c_t = \frac{\gamma_t}{\sum_{i=0}^{t} \gamma_i}. \tag{18}$$

Initializing $\boldsymbol{z}_{-1} = \boldsymbol{x}_0$, we then have

$$\sum_{t=0}^{T} \gamma_t (f(\boldsymbol{x}_T) - f(\boldsymbol{x}_\star)) \leq \gamma_0 (f(\boldsymbol{x}_0) - f(\boldsymbol{x}_\star)) + \sum_{t=0}^{T} \gamma_t \langle \nabla f(\boldsymbol{y}_t), \boldsymbol{z}_{t-1} - \boldsymbol{x}_\star \rangle. \tag{19}$$

*Proof.* Applying Lemma A.1 and multiplying (14) by $\gamma_{t+1}$,

$$\frac{\gamma_{t+1}}{c_{t+1}} f(\boldsymbol{x}_{t+1}) - \gamma_{t+1}\left(\frac{1}{c_{t+1}} - 1\right) f(\boldsymbol{x}_t) - \gamma_{t+1} f(\boldsymbol{x}_\star) \leq \gamma_{t+1} \langle \nabla f(\boldsymbol{y}_{t+1}), \boldsymbol{z}_t - \boldsymbol{x}_\star \rangle. \tag{20}$$

Summing of the left-hand side of (20) from $t = 0$ to $T - 1$ gives

$$\sum_{t=0}^{T-1} \left( \frac{\gamma_{t+1}}{c_{t+1}} f(\boldsymbol{x}_{t+1}) - \gamma_{t+1} \left( \frac{1}{c_{t+1}} - 1 \right) f(\boldsymbol{x}_t) - \gamma_{t+1} f(\boldsymbol{x}_\star) \right)$$

$$= \frac{\gamma_T}{c_T} f(\boldsymbol{x}_T) - \gamma_1 \left( \frac{1}{c_1} - 1 \right) f(\boldsymbol{x}_0) - \sum_{t=1}^{T} \gamma_t f(\boldsymbol{x}_\star) + \sum_{t=1}^{T-1} \left( \frac{\gamma_t}{c_t} - \gamma_{t+1} \left( \frac{1}{c_{t+1}} - 1 \right) \right) f(\boldsymbol{x}_t).$$
(21)

Using (18) we have that the right most term is zero since

$$\frac{\gamma_t}{c_t} - \gamma_{t+1} \left( \frac{1}{c_{t+1}} - 1 \right) = \sum_{i=0}^{t} \gamma_i - \left( \sum_{i=0}^{t+1} \gamma_i - \gamma_{t+1} \right) = 0.$$

We chose the $c_t$ coefficients given in (18) so that the above would be zero. Indeed, instead of plugging in (18), if we set the above to zero, and unroll the recurrence in $c_t$ we get:

$$\frac{\gamma_{t+1}}{c_{t+1}} = \gamma_{t+1} + \frac{\gamma_t}{c_t}$$

$$= \gamma_{t+1} + \gamma_t + \frac{\gamma_{t-1}}{c_{t-1}}$$

$$= \cdots$$

$$= \sum_{i=1}^{t+1} \gamma_i + \frac{\gamma_0}{c_0},$$

which gives

$$c_{t+1} = \frac{\gamma_{t+1}}{\sum_{i=0}^{t+1} \gamma_i}$$

where we have chosen $c_0 = 1$. Thus we arrive at the same recurrence. Similarly,

$$\gamma_1 \left( \frac{1}{c_1} - 1 \right) = \frac{\gamma_0}{c_0} = \gamma_0$$

and

$$\frac{\gamma_T}{c_T} = \sum_{i=0}^{T} \gamma_i.$$

Consequently (21) can be written as

$$\sum_{t=0}^{T-1} \frac{\gamma_{t+1}}{c_{t+1}} f(\boldsymbol{x}_{t+1}) - \gamma_{t+1} \left( \frac{1}{c_{t+1}} - 1 \right) f(\boldsymbol{x}_t) - \gamma_{t+1} f(\boldsymbol{x}_\star)$$

$$= \sum_{t=0}^{T} \gamma_t f(\boldsymbol{x}_T) - \gamma_0 f(\boldsymbol{x}_0) - \sum_{t=1}^{T} \gamma_t f(\boldsymbol{x}_\star)$$

$$= \sum_{t=0}^{T} \gamma_t (f(\boldsymbol{x}_T) - f(\boldsymbol{x}_\star)) - \gamma_0 (f(\boldsymbol{x}_0) - f(\boldsymbol{x}_\star)).$$
(22)

Putting this back to (20), we can write

$$\sum_{t=1}^{T} \gamma_t (f(\boldsymbol{x}_T) - f(\boldsymbol{x}_\star)) \leq \gamma_0 (f(\boldsymbol{x}_0) - f(\boldsymbol{x}_\star)) + \sum_{t=0}^{T-1} \gamma_{t+1} \langle \nabla f(\boldsymbol{y}_{t+1}), \boldsymbol{z}_t - \boldsymbol{x}_\star \rangle$$

$$= \gamma_0 (f(\boldsymbol{x}_0) - f(\boldsymbol{x}_\star)) + \sum_{t=1}^{T} \gamma_t \langle \nabla f(\boldsymbol{y}_t), \boldsymbol{z}_{t-1} - \boldsymbol{x}_\star \rangle.$$

Because of the convexity of $f$, we know that

$$\langle \nabla f(\boldsymbol{x}_0), \boldsymbol{x}_0 - \boldsymbol{x}_\star \rangle \geq 0.$$

Also because we initialize $z_{-1} = x_0$, we have

$$\langle \nabla f(y_0), z_{-1} - x_\star \rangle \geq 0.$$

Therefore, with $\gamma_0 \geq 0$, we obtain

$$\sum_{t=1}^{T} \gamma_t (f(x_T) - f(x_\star)) \leq \gamma_0 (f(x_0) - f(x_\star)) + \sum_{t=0}^{T} \gamma_t \langle \nabla f(y_t), z_{t-1} - x_\star \rangle.$$

□

## A.2 PROOF OF THEOREM 2.1

**Theorem 2.1.** Let $f \colon \mathbb{R}^d \to \mathbb{R}$ be convex and $G$-Lipschitz continuous. Let $\{x_t, y_t, z_t\}$ be generated from (1), (2), (3). Suppose that

$$c_t = \frac{\gamma_t}{\sum_{i=0}^{t} \gamma_i} \qquad (6)$$

for $t = 1, \dots, T$. Initializing $z_{-1} = x_0$, we then have

$$\mathbb{E}\left[f(x_T) - f(x_\star)\right] \leq \frac{\frac{1}{2}\|x_0 - x_\star\|^2 + \gamma_0(f(x_0) - f(x_\star))}{\sum_{t=0}^{T} \gamma_t} + \sum_{t=0}^{T} \frac{\frac{1}{2}\gamma_t^2 G^2}{\sum_{t=0}^{T} \gamma_t}. \qquad (7)$$

*Proof.* Having Lemma A.2 established, it remains to bound the last term of (19). Write $g_t = \nabla f(y_t, \zeta_t)$. Using the updating rule (2), we see that, for $t = 0, 1, \dots, T-1$,

$$\|z_t - x_\star\|^2 = \|z_{t-1} - \gamma_t g_t - x_\star\|^2$$
$$= \|z_{t-1} - x_\star\|^2 - 2\gamma_t \langle g_t, z_{t-1} - x_\star \rangle + \gamma_t^2 \|g_t\|^2.$$

Rearranging terms, we have

$$\langle g_t, z_{t-1} - x_\star \rangle = \frac{1}{2\gamma_t}\|z_{t-1} - x_\star\|^2 - \frac{1}{2\gamma_t}\|z_t - x_\star\|^2 + \frac{\gamma_t}{2}\|g_t\|^2. \qquad (23)$$

Taking expectation conditioned on $z_{t-1}$, and noting that $\mathbb{E}_{t-1}[g_t] = \nabla f(y_t)$ gives

$$\langle \nabla f(y_t), z_{t-1} - x_\star \rangle = \frac{1}{2\gamma_t}\|z_{t-1} - x_\star\|^2 - \frac{1}{2\gamma_t}\mathbb{E}_{t-1}\left[\|z_t - x_\star\|^2\right] + \frac{\gamma_t}{2}\mathbb{E}_{t-1}\left[\|g_t\|^2\right]. \quad (24)$$

Taking full expectation and using the law of total expectation gives

$$\mathbb{E}\langle \nabla f(y_t), z_{t-1} - x_\star \rangle = \frac{1}{2\gamma_t}\mathbb{E}\|z_{t-1} - x_\star\|^2 - \frac{1}{2\gamma_t}\mathbb{E}\|z_t - x_\star\|^2 + \frac{\gamma_t}{2}\mathbb{E}\|g_t\|^2.$$

Multiplying by $\gamma_t$ and summing it up from $t = 0$ to $T$, we have

$$\sum_{t=0}^{T} \gamma_t \mathbb{E}\langle \nabla f(y_t), z_{t-1} - x_\star \rangle = \sum_{t=0}^{T} \left( \frac{1}{2}\mathbb{E}\|z_{t-1} - x_\star\|^2 - \frac{1}{2}\mathbb{E}\|z_t - x_\star\|^2 + \frac{\gamma_t^2}{2}\mathbb{E}\|g_t\|^2 \right)$$

$$= \frac{1}{2}\|z_{-1} - x_\star\|^2 - \frac{1}{2}\mathbb{E}\|z_T - x_\star\|^2 + \sum_{t=0}^{T} \frac{\gamma_t^2}{2}\mathbb{E}\|g_t\|^2$$

$$= \frac{1}{2}\|x_0 - x_\star\|^2 - \frac{1}{2}\mathbb{E}\|z_T - x_\star\|^2 + \sum_{t=0}^{T} \frac{\gamma_t^2}{2}\mathbb{E}\|g_t\|^2. \qquad (25)$$

Dropping the negative $-\frac{1}{2}\mathbb{E}\|z_T - x_\star\|^2$ term, and using the above in (19) we have that

$$\sum_{t=0}^{T} \gamma_t \mathbb{E}\left[f(x_T) - f(x_\star)\right] \leq \gamma_0(f(x_0) - f(x_\star)) + \sum_{t=0}^{T} \gamma_t \mathbb{E}\langle \nabla f(y_t), z_{t-1} - x_\star \rangle$$

$$\leq \gamma_0(f(x_0) - f(x_\star)) + \frac{1}{2}\|x_0 - x_\star\|^2 + \sum_{t=0}^{T} \frac{\gamma_t^2}{2}\mathbb{E}\|g_t\|^2.$$

Finally dividing through by $\sum_{t=0}^{T} \gamma_t$ gives the result. □

### A.3   PROOF OF LEMMA 2.2

**Lemma 2.2.** Let $0 \leq T_w \leq T_c \leq T$ and $\gamma > 0$. Suppose that $\{\eta_t\}_{t=0}^T$ follows the `wsd` schedule given in (8). We can determine $\{c_t\}_{t=0}^T$ by

$$c_t = \begin{cases} \frac{2}{t+2}, & \text{if } 0 \leq t \leq T_w, \\ \frac{2}{2t-T_w+2}, & \text{if } T_w < t \leq T_c, \\ \frac{2(T-t+1)}{(T-T_c+1)(2T_c-T_w+2)+(2T-T_c-t+1)(t-T_c)}, & \text{if } T_c < t \leq T. \end{cases} \quad (9)$$

*Proof.* Recall from (6) is given by

$$c_t = \frac{\eta_t}{\sum_{i=0}^t \eta_i}. \quad (26)$$

for some scheduler $\{\eta_t\}_{t=0}^T$. Now, we are ready to obtain $\{c_t\}_{t=0}^T$ by substituting the `wsd` scheduler $\{\eta_t\}_{t=0}^T$ and applying the arithmetic formula. Specifically, for $0 \leq t \leq T_w$,

$$c_t = \frac{\frac{t+1}{T_w+1}}{\frac{\sum_{i=0}^t (i+1)}{T_w+1}} = \frac{t+1}{\frac{(t+1)(t+2)}{2}} = \frac{2}{t+2}.$$

Since

$$\sum_{i=0}^{T_w} \eta_i = \sum_{i=0}^{T_w} \frac{t+1}{T_w+1} = \frac{\frac{(T_w+1)(T_w+2)}{2}}{T_w+1} = \frac{T_w+2}{2}, \quad (27)$$

we obtain, for $T_w < t \leq T_c$,

$$c_t = \frac{1}{\sum_{i=0}^{T_w} \eta_i + \sum_{i=T_w+1}^t \eta_i} = \frac{1}{\frac{T_w+2}{2} + (t-T_w)} = \frac{2}{2t-T_w+2}.$$

Applying (27) again and using

$$\sum_{i=T_w+1}^{T_c} \eta_i = T_c - T_w,$$

we also have, for $T_c < t \leq T$,

$$c_t = \frac{\frac{T-t+1}{T-T_c+1}}{\sum_{i=0}^{T_w} \eta_i + \sum_{i=T_w+1}^{T_c} \eta_i + \sum_{i=T_c+1}^t \frac{T-i+1}{T-T_c+1}}$$

$$= \frac{\frac{T-t+1}{T-T_c+1}}{\frac{T_w+2}{2} + (T_c - T_w) + \frac{(2T-T_c-t+1)(t-T_c)}{2(T-T_c+1)}}$$

$$= \frac{2(T-t+1)}{(T-T_c+1)(2T_c-T_w+2)+(2T-T_c-t+1)(t-T_c)}.$$

$\square$

### A.4   PROOF OF COROLLARY 2.3

**Corollary 2.3.** Let $D = \|x_0 - x_\star\|$. Using the `wsd` parameters $(c_t, \eta_t)$ given in (9) and (8), with a base learning rate of $\gamma = \frac{D}{G\sqrt{\sum_{t=0}^T \eta_t^2}}$, we have the convergence

$$\mathbb{E}\left[f(x_T) - \inf f\right] \leq \frac{2\eta_0(f(x_0) - f(x_\star))}{T + T_c - T_w + 2} + \frac{2\sqrt{\frac{2}{3}}DG}{\sqrt{T + T_c - T_w + 2}} \simeq \mathcal{O}\left(\frac{DG}{\sqrt{T}}\right). \quad (10)$$

*Proof.* Using the arithmetic sum formula, we can write

$$\sum_{t=0}^{T_w-1} \eta_t = \frac{\sum_{t=0}^{T_w-1}(t+1)}{T_w+1} = \frac{T_w + \sum_{t=0}^{T_w-1} t}{T_w+1} = \frac{T_w + \frac{T_w(T_w-1)}{2}}{T_w+1} = \frac{T_w}{2};$$

$$\sum_{t=T_w}^{T_c-1} \eta_t = T_c - 1 - T_w + 1 = T_c - T_w$$

$$\sum_{t=T_c}^{T} \eta_t = \sum_{t=T_c}^{T} \frac{T-t+1}{T-T_c+1} = \frac{(T+1)(T-T_c+1)}{T-T_c+1} - \frac{\sum_{t=T_c}^{T} t}{T-T_c+1}$$

$$= T+1 - \frac{(T_c+T)(T-T_c+1)}{2(T-T_c+1)} = T+1 - \frac{T_c+T}{2}. \tag{28}$$

Combining, we have

$$\sum_{t=0}^{T} \eta_t = \frac{T_w}{2} + T_c - T_w + T + 1 - \frac{T_c+T}{2} = \frac{T+T_c-T_w+2}{2}. \tag{29}$$

Also, using the fact that

$$\sum_{k=1}^{n} k^2 = \frac{n(n+1)(2n+1)}{6}, \tag{30}$$

we can compute

$$\sum_{t=0}^{T_w-1} \eta_t^2 = \sum_{t=0}^{T_w-1} \frac{(t+1)^2}{(T_w+1)^2} = \frac{\sum_{t=0}^{T_w-1} t^2 + 2\sum_{t=0}^{T_w-1} t + T_w}{(T_w+1)^2}$$

$$= \frac{\frac{T_w(T_w-1)(2T_w-1)}{6} + 2\frac{T_w(T_w-1)}{2} + T_w}{(T_w+1)^2} = \frac{T_w}{T_w+1} \cdot \frac{\frac{(T_w-1)(2T_w-1)}{6} + T_w}{T_w+1}$$

$$\leq \frac{2T_w^2 + 3T_w + 1}{6(T_w+1)} = \frac{(2T_w+1)(T_w+1)}{6(T_w+1)} = \frac{2T_w+1}{6};$$

$$\sum_{t=T_w}^{T_c-1} \eta_t^2 = T_c - 1 - T_w + 1 = T_c - T_w;$$

$$\sum_{t=T_c}^{T} \eta_t^2 = \frac{\sum_{t=T_c}^{T}(T-t+1)^2}{(T-T_c+1)^2} = \frac{\sum_{t=1}^{T-T_c+1} t^2}{(T-T_c+1)^2}$$

$$= \frac{(T-T_c+1)(T-T_c+2)(2T-2T_c+3)}{6(T-T_c+1)^2}$$

$$= \frac{(T-T_c+2)(2T-2T_c+3)}{6(T-T_c+1)}$$

$$= \frac{(T-T_c+1)(2T-2T_c+3) + (2T-2T_c+3)}{6(T-T_c+1)}$$

$$= \frac{1}{6}\left(2T - 2T_c + 3 + \frac{2T-2T_c+3}{T-T_c+1}\right)$$

$$= \frac{1}{6}\left(2T - 2T_c + 3 + \frac{2(T-T_c+1)+1}{T-T_c+1}\right)$$

$$= \frac{1}{6}\left(2T - 2T_c + 3 + 2 + \frac{1}{T-T_c+1}\right)$$

$$\leq \frac{1}{6}(2T - 2T_c + 3 + 2 + 1) = \frac{T}{3} - \frac{T_c}{3} + 1. \tag{31}$$

Combining, we have

$$\sum_{t=0}^{T} \eta_t^2 = \frac{2T_w + 1}{6} + T_c - T_w + \frac{T}{3} - \frac{T_c}{3} + 1 = \frac{T + 2T_c - 2T_w}{3} + \frac{7}{6}$$

$$\leq \frac{2}{3}(T + T_c - T_w + 2). \tag{32}$$

Applying the results to Theorem 2.1 then establishes the corollary. $\qquad\square$

### A.5 Comments on the Weights $c_t$ in Defazio et al. (2024)

Defazio et al. (2024) suggested the convergence rate of $\mathcal{O}(1/\sqrt{T})$ as long as the averaging parameter is in the form of $c_t = w_t / \sum_{i=1}^{t} w_i$ for any $w_t \in [0, 1]$ for $t = 1, \ldots, T$. While the condition might look slightly more general than our proposed $c_t$ in (6), we show that, after applying the standard online convex optimization technique to (Defazio et al., 2024, Theorem 2), the averaging parameter $c_t$ has to satisfy (6) in order to get a valid convergence bound.

To show this, let us first recall (Defazio et al., 2024, Theorem 2).

**Theorem A.3** (Defazio et al. (2024, Theorem 2)). *Let $f \colon \mathbb{R}^d \to \mathbb{R}$ be a convex function and $\boldsymbol{\zeta}_1, \ldots, \boldsymbol{\zeta}_T$ be an iid sequence. Let $\beta_1, \ldots, \beta_T$ and $w_1, \ldots, w_T$ be numbers in $[0, 1]$ that are independent of $\boldsymbol{\zeta}_1, \ldots, \boldsymbol{\zeta}_T$. Consider the iterates $(\boldsymbol{x}_t, \boldsymbol{y}_t, \boldsymbol{z}_t)$ generated by the following:*

$$\boldsymbol{x}_t = \underbrace{\left(1 - \frac{w_t}{\sum_{i=1}^{t} w_i}\right)}_{=:1-c_t} \boldsymbol{x}_{t-1} + \underbrace{\frac{w_t}{\sum_{i=1}^{t} w_i}}_{=:c_t} \boldsymbol{z}_t \tag{33}$$

$$\boldsymbol{y}_t = \beta_t \boldsymbol{x}_t + (1 - \beta_t)\boldsymbol{z}_t \tag{34}$$

$$\boldsymbol{z}_{t+1} = \boldsymbol{z}_t - \gamma_t \boldsymbol{g}_t, \ \boldsymbol{g}_t := \nabla f(\boldsymbol{y}_t, \boldsymbol{\zeta}_t). \tag{35}$$

*Then, we have that*

$$\mathbb{E}\left[f(\boldsymbol{x}_T) - f(\boldsymbol{x}_\star)\right] \leq \frac{\mathbb{E}\left[\sum_{t=1}^{T} w_t \langle \boldsymbol{g}_t, \boldsymbol{z}_t - \boldsymbol{x}_\star \rangle\right]}{\sum_{i=1}^{T} w_i}.$$

Before going into the proof, we would like to give a heads-up that the indices of $\boldsymbol{z}_t$ in Defazio et al. (2024) (as shown in (33)–(35)) is slightly different from our paper (given in (1)–(3)). In this part, we will stick to the updating rule (33)–(35) to derive the condition on $c_t$ based on the results in Defazio et al. (2024, Theorem 2).

From the updating rule (35), we know that

$$\|\boldsymbol{z}_{t+1} - \boldsymbol{x}_\star\|^2 = \|\boldsymbol{z}_t - \gamma_t \boldsymbol{g}_t - \boldsymbol{x}_\star\|^2$$
$$= \|\boldsymbol{z}_t - \boldsymbol{x}_\star\|^2 - 2\gamma_t \langle \boldsymbol{g}_t, \boldsymbol{z}_t - \boldsymbol{x}_\star \rangle + \gamma_t^2 \|\boldsymbol{g}_t\|^2,$$

which implies

$$\langle \boldsymbol{g}_t, \boldsymbol{z}_t - \boldsymbol{x}_\star \rangle = \frac{1}{2\gamma_t}\|\boldsymbol{z}_t - \boldsymbol{x}_\star\|^2 - \frac{1}{2\gamma_t}\|\boldsymbol{z}_{t+1} - \boldsymbol{x}_\star\|^2 + \frac{\gamma_t}{2}\|\boldsymbol{g}_t\|^2.$$

Therefore, multiplying by $w_t$, taking expectation, and summing it up from $t = 1$ to $T$ would yield

$$\mathbb{E}\left[\sum_{t=1}^{T} w_t \langle \boldsymbol{g}_t, \boldsymbol{z}_t - \boldsymbol{x}_\star \rangle\right] = \mathbb{E}\left[\sum_{t=1}^{T} \left(\frac{w_t}{2\gamma_t}\|\boldsymbol{z}_t - \boldsymbol{x}_\star\|^2 - \frac{w_t}{2\gamma_t}\|\boldsymbol{z}_{t+1} - \boldsymbol{x}_\star\|^2 + \frac{w_t \gamma_t}{2}\|\boldsymbol{g}_t\|^2\right)\right]$$

$$\leq \frac{w_1}{2\gamma_1}\|\boldsymbol{z}_1 - \boldsymbol{x}_\star\|^2 + \sum_{t=2}^{T} \mathbb{E}\left[\left(\frac{w_t}{2\gamma_t} - \frac{w_{t-1}}{2\gamma_{t-1}}\right)\|\boldsymbol{z}_t - \boldsymbol{x}_\star\|^2\right] + \sum_{t=1}^{T} \frac{w_t \gamma_t}{2}\mathbb{E}\left[\|\boldsymbol{g}_t\|^2\right].$$

Therefore, to obtain a last-iterate convergence bound, we want

$$\frac{w_t}{\gamma_t} = \frac{w_{t-1}}{\gamma_{t-1}} \tag{36}$$

for $t = 2, \ldots, T$ such that the second term gets canceled out. Unrolling,

$$
\begin{aligned}
w_t &= w_{t-1} \cdot \frac{\gamma_t}{\gamma_{t-1}} \\
&= w_{t-2} \cdot \frac{\gamma_{t-1}}{\gamma_{t-2}} \cdot \frac{\gamma_t}{\gamma_{t-1}} = w_{t-2} \cdot \frac{\gamma_t}{\gamma_{t-2}} \\
&= \cdots \\
&= \gamma_t \cdot \frac{w_1}{\gamma_1}.
\end{aligned}
\tag{37}
$$

Therefore, the condition on $c_t$ is given by

$$
c_t = \frac{w_t}{\sum_{i=1}^{t} w_i} = \frac{\gamma_t \cdot \frac{w_1}{\gamma_1}}{\sum_{i=1}^{t} \gamma_i \cdot \frac{w_1}{\gamma_1}} = \frac{\gamma_t}{\sum_{i=1}^{t} \gamma_i},
$$

which is the same as our condition (6). Moreover, using (37) again, we have that

$$
\sum_{i=1}^{T} w_i = \frac{w_1}{\gamma_1} \sum_{i=1}^{T} \gamma_i,
$$

and hence we have the convergence

$$
\begin{aligned}
\mathbb{E}\left[f(\boldsymbol{x}_T) - f(\boldsymbol{x}_\star)\right] &\leq \frac{\mathbb{E}\left[\sum_{t=1}^{T} w_t \langle \boldsymbol{g}_t, \boldsymbol{z}_t - \boldsymbol{x}_\star \rangle\right]}{\sum_{i=1}^{T} w_i} \\
&\leq \frac{\frac{w_1}{2\gamma_1} \|\boldsymbol{z}_1 - \boldsymbol{x}_\star\|^2 + \sum_{t=1}^{T} \frac{w_t \gamma_t}{2} \mathbb{E}\left[\|\boldsymbol{g}_t\|^2\right]}{\sum_{i=1}^{T} w_i} \\
&= \frac{\frac{1}{2} \|\boldsymbol{z}_1 - \boldsymbol{x}_\star\|^2 + \frac{1}{2} \sum_{t=1}^{T} \gamma_t^2 \mathbb{E}\left[\|\boldsymbol{g}_t\|^2\right]}{\sum_{t=1}^{T} \gamma_t},
\end{aligned}
$$

which achieves the same bound as in Theorem 2.1.

# B PROOFS FOR POLYAK STEPSIZE

## B.1 DERIVATION OF THE SCHEDULEP LEARNING RATE

Starting by expanding the squares of the distance to the solution $\boldsymbol{x}_\star$ we have

$$
\|\boldsymbol{z}_t - \boldsymbol{x}_\star\|^2 = \|\boldsymbol{z}_{t-1} - \boldsymbol{x}_\star\|^2 - 2\gamma_t \langle \nabla f(\boldsymbol{y}_t, \boldsymbol{\zeta}_t), \boldsymbol{z}_{t-1} - \boldsymbol{x}_\star \rangle + \gamma_t^2 \|\nabla f(\boldsymbol{y}_t, \boldsymbol{\zeta}_t)\|^2.
\tag{38}
$$

We could now minimize the right hand side in $\gamma_t$, but then the solution would depend directly on the unknown $\boldsymbol{x}_\star$. So before minimizing in $\gamma_t$, we need to upper bound the right hand side with terms we do know.

To simplify notation, let us consider $\beta_t \equiv \beta$ for all $t$. Re-arranging (1) gives

$$
\boldsymbol{z}_{t-1} = \frac{1}{1-\beta} \boldsymbol{y}_t - \left(\frac{1}{1-\beta} - 1\right) \boldsymbol{x}_t = \boldsymbol{y}_t - \frac{\beta}{1-\beta}(\boldsymbol{x}_t - \boldsymbol{y}_t)
\tag{39}
$$

Now the above in (38) gives

$$
\begin{aligned}
\|\boldsymbol{z}_t - \boldsymbol{x}_\star\|^2 = \|\boldsymbol{z}_{t-1} - \boldsymbol{x}_\star\|^2 + \gamma_t^2 \|\nabla f(\boldsymbol{y}_t, \boldsymbol{\zeta}_t)\|^2 \\
- 2\gamma_t \langle \nabla f(\boldsymbol{y}_t, \boldsymbol{\zeta}_t), \boldsymbol{y}_t - \boldsymbol{x}_\star \rangle + 2\gamma_t \frac{\beta}{1-\beta} \langle \nabla f(\boldsymbol{y}_t, \boldsymbol{\zeta}_t), \boldsymbol{x}_t - \boldsymbol{y}_t \rangle.
\end{aligned}
$$

Now using convexity we have that

$$
- \langle \nabla f(\boldsymbol{y}_t, \boldsymbol{\zeta}_t), \boldsymbol{y}_t - \boldsymbol{x}_\star \rangle \leq f_{\boldsymbol{\zeta}_t}(\boldsymbol{x}_\star) - f_{\boldsymbol{\zeta}_t}(\boldsymbol{y}_t)
$$

and using that $\boldsymbol{y}_t - \boldsymbol{x}_t = (1-\beta)(\boldsymbol{z}_{t-1} - \boldsymbol{x}_t)$ gives

$$
\begin{aligned}
\|\boldsymbol{z}_t - \boldsymbol{x}_\star\|^2 \leq \|\boldsymbol{z}_{t-1} - \boldsymbol{x}_\star\|^2 + \gamma_t^2 \|\nabla f(\boldsymbol{y}_t, \boldsymbol{\zeta}_t)\|^2 \\
- 2\gamma_t \left((f_{\boldsymbol{\zeta}_t}(\boldsymbol{y}_t) - f_{\boldsymbol{\zeta}_t}(\boldsymbol{x}_\star)) - 2\gamma_t \beta \langle \nabla f(\boldsymbol{y}_t, \boldsymbol{\zeta}_t), \boldsymbol{z}_{t-1} - \boldsymbol{x}_t \rangle\right).
\end{aligned}
\tag{40}
$$

Minimizing over $\gamma_t \geq 0$ gives

$$
\gamma_t = \frac{\left(f_{\boldsymbol{\zeta}_t}(\boldsymbol{y}_t) - f_{\boldsymbol{\zeta}_t}(\boldsymbol{x}_\star) + \beta \langle \nabla f(\boldsymbol{y}_t, \boldsymbol{\zeta}_t), \boldsymbol{z}_{t-1} - \boldsymbol{x}_t \rangle\right)_+}{\|\nabla f(\boldsymbol{y}_t, \boldsymbol{\zeta}_t)\|^2}.
\tag{41}
$$

B.2 AUXILIARY LEMMAS

**Lemma B.1** (Extended Titu's Lemma). For any random variable $X$ and positive-valued random variable $Y$, it holds

$$\mathbb{E}\left[\frac{(X)_+^2}{Y}\right] \geq \frac{(\mathbb{E}[X])_+^2}{\mathbb{E}[Y]}. \tag{42}$$

In addition, for any numbers $a_0, \dots, a_k$ and positive numbers $b_0, \dots, b_k$, we have

$$\sum_{t=0}^{k} \frac{(a_t)_+^2}{b_t} \geq \frac{\left(\sum_{t=0}^{k} a_t\right)_+^2}{\sum_{t=0}^{k} b_t}. \tag{43}$$

**Lemma B.2.** If $f_\zeta$ is convex for every $\zeta$, and we use the learning rate (41) we have that

$$\|\boldsymbol{z}_t - \boldsymbol{x}_\star\|^2 \leq \|\boldsymbol{z}_{t-1} - \boldsymbol{x}_\star\|^2 - \frac{\left(f_{\boldsymbol{\zeta}_t}(\boldsymbol{y}_t) - f_{\boldsymbol{\zeta}_t}(\boldsymbol{x}_\star) + \beta\left\langle\nabla f(\boldsymbol{y}_t, \boldsymbol{\zeta}_t), \boldsymbol{z}_{t-1} - \boldsymbol{x}_t\right\rangle\right)_+^2}{\|\nabla f(\boldsymbol{y}_t, \boldsymbol{\zeta}_t)\|^2}. \tag{44}$$

As a consequence we also have that $\|\boldsymbol{z}_t - \boldsymbol{x}_\star\|$, $\|\boldsymbol{x}_t - \boldsymbol{x}_\star\|$ and $\|\boldsymbol{y}_t - \boldsymbol{x}_\star\|$ are less than $\|\boldsymbol{x}_0 - \boldsymbol{x}_\star\|$. Furthermore, taking expectation we have that

$$\mathbb{E}\left[\|\boldsymbol{z}_t - \boldsymbol{x}_\star\|^2\right] \leq \mathbb{E}\left[\|\boldsymbol{z}_{t-1} - \boldsymbol{x}_\star\|^2\right] - \frac{\left(\mathbb{E}\left[f(\boldsymbol{y}_t) - f(\boldsymbol{x}_\star) + \beta\left\langle\nabla f(\boldsymbol{y}_t), \boldsymbol{z}_{t-1} - \boldsymbol{x}_t\right\rangle\right]\right)_+^2}{\mathbb{E}\left[\|\nabla f(\boldsymbol{y}_t, \boldsymbol{\zeta}_t)\|^2\right]}. \tag{45}$$

*Proof.* Inserting (41) into (40) gives the first result, which also shows that $\|\boldsymbol{z}_t - \boldsymbol{x}_\star\| \leq \|\boldsymbol{z}_0 - \boldsymbol{x}_\star\| = \|\boldsymbol{x}_0 - \boldsymbol{x}_\star\|$. Since $\boldsymbol{x}_{t+1}$ is a convex combination of $\boldsymbol{x}_t$ and $\boldsymbol{z}_t$, we have that

$$\|\boldsymbol{x}_{t+1} - \boldsymbol{x}_\star\| \leq (1 - c_{t+1})\|\boldsymbol{x}_t - \boldsymbol{x}_\star\| + c_{t+1}\|\boldsymbol{z}_t - \boldsymbol{x}_\star\|$$

from which we can use induction to show $\|\boldsymbol{x}_t - \boldsymbol{x}_\star\| \leq \|\boldsymbol{x}_0 - \boldsymbol{x}_\star\|$. Furthermore, since $\boldsymbol{y}_t$ is a convex combination of $\boldsymbol{z}_{t-1}$ and $\boldsymbol{x}_t$, it also follows by induction that $\|\boldsymbol{y}_t - \boldsymbol{x}_\star\| \leq \|\boldsymbol{x}_0 - \boldsymbol{x}_\star\|$.

Taking conditional expectation over (44) given $\boldsymbol{x}_t$ and $\boldsymbol{z}_{t-1}$ and using Lemma B.1 gives

$$\mathbb{E}_t[\|\boldsymbol{z}_t - \boldsymbol{x}_\star\|^2] \leq \|\boldsymbol{z}_{t-1} - \boldsymbol{x}_\star\|^2 - \frac{\left(f(\boldsymbol{y}_t) - f(\boldsymbol{x}_\star) + \beta\langle\nabla f(\boldsymbol{y}_t), \boldsymbol{z}_{t-1} - \boldsymbol{x}_t\rangle\right)_+^2}{\mathbb{E}_t[\|\nabla f(\boldsymbol{y}_t, \boldsymbol{\zeta}_t)\|^2]}. \tag{46}$$

Finally, taking total expectation over (46), and using the law of total expectation and Lemma B.1 again, yields (45). $\square$

Next we develop the Bregman viewpoint of this method.

**Lemma B.3.** Let $\lambda = \frac{\beta}{1-\beta}$. It follows that

$$\begin{aligned} f(\boldsymbol{y}_t) - f(\boldsymbol{x}_\star) + \beta\left\langle\nabla f(\boldsymbol{y}_t), \boldsymbol{z}_{t-1} - \boldsymbol{x}_t\right\rangle = {} & (1 + \lambda)(f(\boldsymbol{y}_t) - f(\boldsymbol{x}_\star)) \\ & - \lambda(f(\boldsymbol{x}_t) - f(\boldsymbol{x}_\star)) \\ & + \lambda B_f(\boldsymbol{x}_t, \boldsymbol{y}_t), \end{aligned} \tag{47}$$

where $B_f(\boldsymbol{x}_t, \boldsymbol{y}_t)$ is the Bregman divergence of $f$ that is

$$B_f(\boldsymbol{x}, \boldsymbol{y}) := f(\boldsymbol{x}) - f(\boldsymbol{y}) - \left\langle\nabla f(\boldsymbol{y}), \boldsymbol{x} - \boldsymbol{y}\right\rangle.$$

*Proof.* Using $\boldsymbol{z}_{t-1} - \boldsymbol{x}_t = \frac{1}{1-\beta}(\boldsymbol{y}_t - \boldsymbol{x}_t)$ which follows from (1) gives

$$\begin{aligned} f(\boldsymbol{y}_t) - f(\boldsymbol{x}_\star) + \beta\left\langle\nabla f(\boldsymbol{y}_t), \boldsymbol{z}_{t-1} - \boldsymbol{x}_t\right\rangle = {} & f(\boldsymbol{y}_t) - f(\boldsymbol{x}_\star) - \frac{\beta}{1-\beta}\left\langle\nabla f(\boldsymbol{y}_t), \boldsymbol{x}_t - \boldsymbol{y}_t\right\rangle \\ = {} & (1 + \lambda)(f(\boldsymbol{y}_t) - f(\boldsymbol{x}_\star)) - \lambda(f(\boldsymbol{x}_t) - f(\boldsymbol{x}_\star)) \\ & + \lambda\big(f(\boldsymbol{x}_t) - f(\boldsymbol{y}_t) - \left\langle\nabla f(\boldsymbol{y}_t), \boldsymbol{x}_t - \boldsymbol{y}_t\right\rangle\big). \end{aligned}$$

□

**Lemma B.4.** Let $c_t = 1/(t+1)$. Initializing $\boldsymbol{z}_{-1} = \boldsymbol{x}_0$, it follows that

$$\mathbb{E}\left[\|\boldsymbol{z}_t - \boldsymbol{x}_\star\|^2\right] \leq \|\boldsymbol{x}_0 - \boldsymbol{x}_\star\|^2 \tag{48}$$

$$-\frac{\left((t+1)\mathbb{E}\left[f(\boldsymbol{x}_t) - f(\boldsymbol{x}_\star)\right] + \lambda \sum_{k=0}^t \mathbb{E}\left[B_f(\boldsymbol{x}_k, \boldsymbol{y}_k)\right]\right)_+^2}{\sum_{k=0}^t \mathbb{E}\left[\|\nabla f(\boldsymbol{y}_k, \boldsymbol{\zeta}_k)\|^2\right]}.$$

*Proof.* Using (47) in (45) gives

$$\mathbb{E}\left[\|\boldsymbol{z}_t - \boldsymbol{x}_\star\|^2\right] = \mathbb{E}\left[\|\boldsymbol{z}_{t-1} - \boldsymbol{x}_\star\|^2\right]$$

$$-\frac{\left(\mathbb{E}\left[(1+\lambda)(f(\boldsymbol{y}_t) - f(\boldsymbol{x}_\star)) - \lambda(f(\boldsymbol{x}_t) - f(\boldsymbol{x}_\star)) + \lambda B_f(\boldsymbol{x}_t, \boldsymbol{y}_t)\right]\right)_+^2}{\mathbb{E}\left[\|\nabla f(\boldsymbol{y}_t, \boldsymbol{\zeta}_t)\|^2\right]}. \tag{49}$$

Therefore, unrolling (49) gives

$$\mathbb{E}\left[\|\boldsymbol{z}_t - \boldsymbol{x}_\star\|^2\right] \leq \|\boldsymbol{z}_{-1} - \boldsymbol{x}_\star\|^2 - \sum_{k=0}^t \frac{\left(a_k\right)_+^2}{b_k},$$

where we define $a_k := \mathbb{E}\left[(1+\lambda)(f(\boldsymbol{y}_k) - f(\boldsymbol{x}_\star)) - \lambda(f(\boldsymbol{x}_k) - f(\boldsymbol{x}_\star)) + \lambda B_f(\boldsymbol{x}_k, \boldsymbol{y}_k)\right]$ and $b_k = \mathbb{E}\left[\|\nabla f(\boldsymbol{y}_k, \boldsymbol{\zeta}_k)\|^2\right]$. From Lemma B.1, we know that

$$\sum_{k=0}^t \frac{(a_k)_+^2}{b_k} \geq \frac{\left(\sum_{k=0}^t a_k\right)_+^2}{\sum_{k=0}^t b_k}.$$

Therefore, we have that

$$\mathbb{E}\left[\|\boldsymbol{z}_t - \boldsymbol{x}_\star\|^2\right] = \|\boldsymbol{z}_{-1} - \boldsymbol{x}_\star\|^2 \tag{50}$$

$$-\frac{\left(\sum_{k=0}^t \mathbb{E}\left[(1+\lambda)(f(\boldsymbol{y}_k) - f(\boldsymbol{x}_\star)) - \lambda(f(\boldsymbol{x}_k) - f(\boldsymbol{x}_\star)) + \lambda B_f(\boldsymbol{x}_k, \boldsymbol{y}_k)\right]\right)_+^2}{\sum_{k=0}^t \mathbb{E}\left[\|\nabla f(\boldsymbol{y}_k, \boldsymbol{\zeta}_k)\|\right]^2}.$$

To finish the proof of convergence, we need to write $\boldsymbol{y}_t$ as a combination of $\boldsymbol{x}_t$ and $\boldsymbol{x}_{t-1}$ so that we can telescope. To this end note that

$$\boldsymbol{z}_{t-1} = \frac{1}{c_t}\boldsymbol{x}_t + \left(1 - \frac{1}{c_t}\right)\boldsymbol{x}_{t-1}.$$

Substituting this into the $\boldsymbol{y}_t$ update (1) gives

$$\boldsymbol{y}_t = (1-\beta)\left(\frac{1}{c_t}\boldsymbol{x}_t + \left(1 - \frac{1}{c_t}\right)\boldsymbol{x}_{t-1}\right) + \beta\boldsymbol{x}_t$$

$$= \left((1-\beta)\left(\frac{1}{c_t} - 1\right) + 1\right)\boldsymbol{x}_t - (1-\beta)\left(\frac{1}{c_t} - 1\right)\boldsymbol{x}_{t-1}.$$

Let $\rho_t := (1-\beta)\left(\frac{1}{c_t} - 1\right)$. Isolating $\boldsymbol{x}_t$ in the above we have that it can be expressed as a convex combination between $\boldsymbol{y}_t$ and $\boldsymbol{x}_{t-1}$ given by

$$\boldsymbol{x}_t = \frac{1}{1+\rho_t}\boldsymbol{y}_t + \frac{\rho_t}{1+\rho_t}\boldsymbol{x}_{t-1}. \tag{51}$$

Using the convexity of $f$ we have that

$$f(\boldsymbol{x}_t) \leq \frac{1}{1+\rho_t}f(\boldsymbol{y}_t) + \frac{\rho_t}{1+\rho_t}f(\boldsymbol{x}_{t-1}). \tag{52}$$

Re-arranging and isolating $f(\boldsymbol{y}_t)$ gives

$$f(\boldsymbol{y}_t) \geq (1+\rho_t)f(\boldsymbol{x}_t) - \rho_t f(\boldsymbol{x}_{t-1}). \tag{53}$$

Using the above we have that

$$(1 + \lambda)(f(\boldsymbol{y}_t) - f(\boldsymbol{x}_\star)) - \lambda(f(\boldsymbol{x}_t) - f(\boldsymbol{x}_\star)) \geq (1 + \lambda)(1 + \rho_t)(f(\boldsymbol{x}_t) - f(\boldsymbol{x}_\star))$$
$$- (1 + \lambda)\rho_t(f(\boldsymbol{x}_{t-1}) - f(\boldsymbol{x}_\star)))$$
$$- \lambda(f(\boldsymbol{x}_t) - f(\boldsymbol{x}_\star))$$
$$= (1 + (1 + \lambda)\rho_t)(f(\boldsymbol{x}_t) - f(\boldsymbol{x}_\star))$$
$$- (1 + \lambda)\rho_t(f(\boldsymbol{x}_{t-1}) - f(\boldsymbol{x}_\star)))$$

Substituting back $\rho_t := (1 - \beta)\left(\frac{1}{c_t} - 1\right)$ and $1 + \lambda = \frac{1}{1-\beta}$ in the above and using that $c_t = 1/(t + \frac{1}{c_0})$ gives

$$(1 + \lambda)(f(\boldsymbol{y}_t) - f(\boldsymbol{x}_\star)) - \lambda(f(\boldsymbol{x}_t) - f(\boldsymbol{x}_\star))$$
$$\geq \left(\frac{1}{c_t}\right)(f(\boldsymbol{x}_t) - f(\boldsymbol{x}_\star)) - \left(\frac{1}{c_t} - 1\right)(f(\boldsymbol{x}_{t-1}) - f(\boldsymbol{x}_\star))$$
$$= \left(t + \frac{1}{c_0}\right)(f(\boldsymbol{x}_t) - f(\boldsymbol{x}_\star)) - \left(t - 1 + \frac{1}{c_0}\right)(f(\boldsymbol{x}_{t-1}) - f(\boldsymbol{x}_\star)).$$

Using the above we have that

$$\sum_{k=0}^{t} \left((1 + \lambda)(f(\boldsymbol{y}_k) - f(\boldsymbol{x}_\star)) - \lambda(f(\boldsymbol{x}_k) - f(\boldsymbol{x}_\star))\right)$$

$$\geq f(\boldsymbol{x}_0) - f(\boldsymbol{x}_\star) + \sum_{k=1}^{t} \left(\left(k + \frac{1}{c_0}\right)(f(\boldsymbol{x}_k) - f(\boldsymbol{x}_\star)) - \left(k - 1 + \frac{1}{c_0}\right)(f(\boldsymbol{x}_{k-1}) - f(\boldsymbol{x}_\star))\right)$$

$$= f(\boldsymbol{x}_0) - f(\boldsymbol{x}_\star) + \left(t + \frac{1}{c_0}\right)(f(\boldsymbol{x}_t) - f(\boldsymbol{x}_\star)) - \frac{1}{c_0}(f(\boldsymbol{x}_0) - f(\boldsymbol{x}_\star))$$

$$= (t + 1)(f(\boldsymbol{x}_t) - f(\boldsymbol{x}_\star)).$$

Inserting this in (50), together with the monotonicity of the positive part and the initialization that $\boldsymbol{z}_{-1} = \boldsymbol{x}_0$, gives

$$\mathbb{E}\left[\|\boldsymbol{z}_t - \boldsymbol{x}_\star\|^2\right] = \|\boldsymbol{x}_0 - \boldsymbol{x}_\star\|^2 \tag{54}$$
$$- \frac{\left((t + 1)\mathbb{E}\left[f(\boldsymbol{x}_t) - f(\boldsymbol{x}_\star)\right] + \lambda \sum_{k=0}^{t} \mathbb{E}\left[B_f(\boldsymbol{x}_k, \boldsymbol{y}_k)\right]\right)_+^2}{\sum_{k=0}^{t} \mathbb{E}\left[\|\nabla f(\boldsymbol{y}_k, \boldsymbol{\zeta}_k)\|^2\right]}.$$

$\square$

## B.3 Proof of Theorem 3.2

**Theorem 3.2.** Consider the iterates of Algorithm 1 with $c_t = 1/(t + 1)$, $\beta \in [0, 1)$ and $\gamma_{\max} = \infty$. Let $f_{\boldsymbol{\zeta}} : \mathbb{R}^d \to \mathbb{R}$ be a convex function for every $\boldsymbol{\zeta}$. Let

$$B := \{\boldsymbol{x} : \|\boldsymbol{x} - \boldsymbol{x}_\star\| \leq \|\boldsymbol{x}_0 - \boldsymbol{x}_\star\|\} \subset \mathbb{R}^d, \tag{11}$$
$$G^2 := \max_{\boldsymbol{x} \in B} \mathbb{E}_{\boldsymbol{\zeta}} \|\nabla f(\boldsymbol{x}, \boldsymbol{\zeta})\|^2. \tag{12}$$

With the initialization $\boldsymbol{z}_{-1} = \boldsymbol{x}_0$, the suboptimality gap of the *last iterate* $\boldsymbol{x}_t$ converges at a $1/\sqrt{t}$ rate according to

$$\mathbb{E}\left[f(\boldsymbol{x}_t) - f(\boldsymbol{x}_\star)\right] \leq \frac{G\|\boldsymbol{x}_0 - \boldsymbol{x}_\star\|}{\sqrt{t + 1}}. \tag{13}$$

*Proof.* Since $\|\boldsymbol{y}_k - \boldsymbol{x}_\star\| \leq \|\boldsymbol{x}_0 - \boldsymbol{x}_\star\|$ we have that $\mathbb{E}\left[\|\nabla f(\boldsymbol{y}_k, \boldsymbol{\zeta}_k)\|^2\right] \leq G^2$ and re-arranging (48) gives

$$\left((t + 1)\mathbb{E}\left[f(\boldsymbol{x}_t) - f(\boldsymbol{x}_\star)\right] + \lambda \sum_{k=0}^{t} \mathbb{E}\left[B_f(\boldsymbol{x}_k, \boldsymbol{y}_k)\right]\right)_+^2 \leq G^2(t + 1)(\|\boldsymbol{x}_0 - \boldsymbol{x}_\star\|^2 - \|\boldsymbol{z}_t - \boldsymbol{x}_\star\|^2)$$
$$\leq G^2(t + 1)\|\boldsymbol{x}_0 - \boldsymbol{x}_\star\|^2.$$

Since the term on the left is always positive we can drop the positive part, taking square roots, and dividing through by $t + 1$ gives

$$\mathbb{E}\left[f(\boldsymbol{x}_t) - f(\boldsymbol{x}_\star)\right] + \frac{\lambda}{t+1}\sum_{k=0}^{t}\mathbb{E}\left[B_f(\boldsymbol{x}_k, \boldsymbol{y}_k)\right] \leq \frac{G\|\boldsymbol{x}_0 - \boldsymbol{x}_\star\|}{\sqrt{t+1}}.$$

Inserting back $\lambda = \beta/(1 - \beta)$ gives

$$\mathbb{E}\left[f(\boldsymbol{x}_t) - f(\boldsymbol{x}_\star)\right] \leq \mathbb{E}\left[f(\boldsymbol{x}_t) - f(\boldsymbol{x}_\star)\right] + \frac{1}{t+1}\frac{\beta}{1-\beta}\sum_{k=0}^{t}\mathbb{E}\left[B_f(\boldsymbol{x}_k, \boldsymbol{y}_k)\right] \leq \frac{G\|\boldsymbol{x}_0 - \boldsymbol{x}_\star\|}{\sqrt{t+1}}.$$

Finally, we can drop the positive terms given by the Bregman divergences $\mathbb{E}\left[B_f(\boldsymbol{x}_k, \boldsymbol{y}_k)\right]$, giving the final desired result.

$\square$

## C  PRACTICAL & ADAM VERSIONS OF SCHEDULEP

We can also develop a version of `Schedulep` that makes use of any preconditioner, such as the Adam preconditioner.

To derive a preconditioned version of `Schedulep`, let $\boldsymbol{D}_t \in \mathbb{R}^{d\times d}$ be our positive definite symmetric preconditioner, and let $\|\boldsymbol{z}\|_{\boldsymbol{D}_t}^2 := \langle \boldsymbol{D}_t\boldsymbol{z}, \boldsymbol{z}\rangle$ be the norm induced by this preconditioner. The preconditioned version of Schedulefree is given by

$$\boldsymbol{y}_t = (1 - \beta)\boldsymbol{z}_{t-1} + \beta\boldsymbol{x}_t \tag{55}$$

$$\boldsymbol{z}_t = \boldsymbol{z}_{t-1} - \gamma_t\boldsymbol{D}_t^{-1}\nabla f(\boldsymbol{y}_t, \boldsymbol{\zeta}_t) \tag{56}$$

$$\boldsymbol{x}_{t+1} = (1 - c_{t+1})\boldsymbol{x}_t + c_{t+1}\boldsymbol{z}_t \tag{57}$$

We can again upper bound the distance between $\boldsymbol{z}_t$ and a solution $\boldsymbol{x}_\star$, but now under the preconditioned norm via

$$\|\boldsymbol{z}_t - \boldsymbol{x}_\star\|_{\boldsymbol{D}_t}^2 = \|\boldsymbol{z}_{t-1} - \boldsymbol{x}_\star\|_{\boldsymbol{D}_t}^2 - 2\gamma_t\left\langle \boldsymbol{D}_t^{-1}\nabla f(\boldsymbol{y}_t, \boldsymbol{\zeta}_t), \boldsymbol{z}_{t-1} - \boldsymbol{x}_\star\right\rangle_{\boldsymbol{D}_t} + \gamma_t^2\|\nabla f(\boldsymbol{y}_t, \boldsymbol{\zeta}_t)\|_{\boldsymbol{D}_t^{-1}}^2$$

$$= \|\boldsymbol{z}_{t-1} - \boldsymbol{x}_\star\|_{\boldsymbol{D}_t}^2 - 2\gamma_t\left\langle \nabla f(\boldsymbol{y}_t, \boldsymbol{\zeta}_t), \boldsymbol{z}_{t-1} - \boldsymbol{x}_\star\right\rangle + \gamma_t^2\|\nabla f(\boldsymbol{y}_t, \boldsymbol{\zeta}_t)\|_{\boldsymbol{D}_t^{-1}}^2. \tag{58}$$

It only remains to bound the linear term $\langle \nabla f(\boldsymbol{y}_t), \boldsymbol{z}_{t-1} - \boldsymbol{x}_\star\rangle$ for which we follow the exact same steps between (39) and (40) giving

$$\|\boldsymbol{z}_t - \boldsymbol{x}_\star\|_{\boldsymbol{D}_t}^2 \leq \|\boldsymbol{z}_{t-1} - \boldsymbol{x}_\star\|_{\boldsymbol{D}_t}^2 + \gamma_t^2\|\nabla f(\boldsymbol{y}_t, \boldsymbol{\zeta}_t)\|_{\boldsymbol{D}_t^{-1}}^2 \tag{59}$$

$$- 2\gamma_t\left((f_{\boldsymbol{\zeta}_t}(\boldsymbol{y}_t) - f_{\boldsymbol{\zeta}_t}(\boldsymbol{x}_\star)) - 2\beta\gamma_t\left\langle\nabla f(\boldsymbol{y}_t, \boldsymbol{\zeta}_t), \boldsymbol{z}_{t-1} - \boldsymbol{x}_t\right\rangle\right).$$

Minimizing the above in $\gamma_t \geq 0$ gives

$$\gamma_t = \frac{\left(f_{\boldsymbol{\zeta}_t}(\boldsymbol{y}_t) - f_{\boldsymbol{\zeta}_t}(\boldsymbol{x}_\star) + \beta\left\langle\nabla f(\boldsymbol{y}_t, \boldsymbol{\zeta}_t), \boldsymbol{z}_{t-1} - \boldsymbol{x}_t\right\rangle\right)_+}{\|\nabla f(\boldsymbol{y}_t, \boldsymbol{\zeta}_t)\|_{\boldsymbol{D}_t^{-1}}^2}. \tag{60}$$

See Algorithm 2 for the complete pseudo-code.

**Remark C.1** (Practical version). In our code we use a slightly different form given by

$$\gamma_t = \frac{\left(f_{\boldsymbol{\zeta}_t}(\boldsymbol{y}_t) - f_{\boldsymbol{\zeta}_t}(\boldsymbol{x}_\star) + \left\langle\nabla f(\boldsymbol{y}_t, \boldsymbol{\zeta}_t), \boldsymbol{z}_{t-1} - \boldsymbol{y}_t\right\rangle\right)_+}{\|\nabla f(\boldsymbol{y}_t, \boldsymbol{\zeta}_t)\|^2}. \tag{61}$$

This follows from (41) by using that

$$\boldsymbol{x}_t = \frac{1}{\beta}\boldsymbol{y}_t + \left(1 - \frac{1}{\beta}\right)\boldsymbol{z}_{t-1}$$

thus

$$\boldsymbol{z}_{t-1} - \boldsymbol{x}_t = \frac{1}{\beta}\boldsymbol{z}_{t-1} - \frac{1}{\beta}\boldsymbol{y}_t.$$

---

**Algorithm 2** `Adam-Schedulep`: Adam Schedule Free Polyak

---

1: **Input:** $z_{-1} = x_0 \in \mathbb{R}^d$, $\beta \in [0, 1]$, $c_t > 0$
2: **for** $t = 0$ to $T - 1$ **do**
3:     $y_t = (1 - \beta)z_t + \beta x_t$
4:     $\gamma_t = \dfrac{[f_{\zeta_t}(y_t) - f_{\zeta_t}(x_\star) + \beta \langle \nabla f(y_t, \zeta_t), z_t - x_t \rangle]_+}{\|\nabla f(y_t, \zeta_t)\|_{D_t}^2}$
5:     $z_{t+1} = z_t - \gamma_t D_t^{-1} \nabla f(y_t, \zeta_t)$
6:     $x_{t+1} = (1 - c_{t+1})x_t + c_{t+1} z_{t+1}$
7: **end for**
8: **Return:** $x_T$

---

Thus finally
$$\beta \langle \nabla f(y_t, \zeta_t), z_{t-1} - x_t \rangle = \langle \nabla f(y_t, \zeta_t), z_{t-1} - y_t \rangle.$$

## D   Implications to Momentum Method

Since primal averaging is a special case of `schedule-free` when $\beta = 1$, and primal averaging itself is equivalent to `momentum`, our convergence theory for the `schedule-free` method includes `Momentum` as a special case. For example, the last-iterate convergence result in Corollary 2.3 applies to the primal averaging method when $\beta = 1$. This is interesting because of the equivalence between the primal averaging and `momentum`.

---

**Algorithm 3** `Momentum`

---

1: **Input:** $x_0 \in \mathbb{R}^d$, $m_{-1} = 0$, $\alpha_t \geq 0$, $\lambda_t \geq 0$.
2: **for** $t = 0$ to $T - 1$ **do**
3:     $m_t = \frac{\lambda_t}{1 + \lambda_t} m_{t-1} + \frac{1}{1 + \lambda_t} \nabla f(x_t, \zeta_t)$
4:     $x_{t+1} = x_t - \alpha_t m_t$
5: **end for**
6: **Return:** $x_T$

---

The equivalence of the momentum method and the primal averaging method is shown in the following lemma.

**Lemma D.1.** If $(x_t)_{t \in \mathbb{N}}$ is generated by the Momentum Algorithm 3 from parameters $(\alpha_t, \lambda_t)$, then it verifies the primal averaging iterates by choosing any parameters $(\gamma_t, c_t)$ satisfying

$$c_1 \gamma_0 = \frac{\alpha_0}{1 + \lambda_0}, \tag{62}$$

and for $t \geq 1$,

$$\alpha_{t-1} \left( \frac{1}{c_t} - 1 \right) \frac{1 + \lambda_t}{\lambda_t} = \frac{\alpha_t}{c_{t+1}}, \quad \text{and} \quad \gamma_t = \frac{\alpha_{t-1}}{\lambda_t} \left( \frac{1}{c_t} - 1 \right). \tag{63}$$

*Proof.* For the primal averaging iterate, since $z_{-1} = x_0$,

$$x_1 = (1 - c_1)x_0 + c_1(x_0 - \gamma_0 \nabla f(x_0, \zeta_0))$$
$$= x_0 - c_1 \gamma_0 \nabla f(x_0, \zeta_0).$$

For the momentum iterate, since $m_{-1} = 0$,

$$x_1 = x_0 - \frac{\alpha_0}{1 + \lambda_0} \nabla f(x_0, \zeta_0).$$

Hence, they are equivalent when

$$c_1 \gamma_0 = \frac{\alpha_0}{1 + \lambda_0}.$$

Suppose that the iterates of primal averaging and the momentum iterate are equivalent at $(t-1)$-st and $t$-th iteration for some $t \geq 1$. Let us show that their iterates at the $(t+1)$-st iteration are the same; i.e.,

$$\boldsymbol{x}_{t+1}^{\text{momentum}} = \boldsymbol{x}_t - \alpha_t \boldsymbol{m}_t = (1 - c_{t+1})\boldsymbol{x}_t + c_{t+1}\boldsymbol{z}_t = \boldsymbol{x}_{t+1}^{\text{PA}},$$

equivalently,

$$\boldsymbol{z}_t = \boldsymbol{x}_t - \frac{\alpha_t}{c_{t+1}}\boldsymbol{m}_t. \tag{64}$$

Indeed, by the induction hypothesis, we have

$$\boldsymbol{z}_{t-1} = \boldsymbol{x}_{t-1} - \frac{\alpha_{t-1}}{c_t}\boldsymbol{m}_{t-1}. \tag{65}$$

By the updating rule of the primal averaging method and (65), we have

$$\begin{aligned}
\boldsymbol{z}_t &= \boldsymbol{z}_{t-1} - \gamma_t \nabla f(\boldsymbol{x}_t, \boldsymbol{\zeta}_t) \\
&= \boldsymbol{x}_{t-1} - \frac{\alpha_{t-1}}{c_t}\boldsymbol{m}_{t-1} - \gamma_t \nabla f(\boldsymbol{x}_t, \boldsymbol{\zeta}_t) \\
&= \boldsymbol{x}_t - \alpha_{t-1}\left(\frac{1}{c_t} - 1\right)\boldsymbol{m}_{t-1} - \gamma_t \nabla f(\boldsymbol{x}_t, \boldsymbol{\zeta}_t) \\
&= \boldsymbol{x}_t - \alpha_{t-1}\left(\frac{1}{c_t} - 1\right)\frac{1 + \lambda_t}{\lambda_t}\boldsymbol{m}_t - \left(\gamma_t - \frac{\alpha_{t-1}}{\lambda_t}\left(\frac{1}{c_t} - 1\right)\right)\nabla f(\boldsymbol{x}_t, \boldsymbol{\zeta}_t).
\end{aligned}$$

The last two lines follow from the updating rule of the momentum method. Hence, we have shown (64) to hold when

$$\alpha_{t-1}\left(\frac{1}{c_t} - 1\right)\frac{1 + \lambda_t}{\lambda_t} = \frac{\alpha_t}{c_{t+1}}, \quad \text{and} \quad \gamma_t = \frac{\alpha_{t-1}}{\lambda_t}\left(\frac{1}{c_t} - 1\right).$$

$\square$

The above lemma shows that, as long as the hyperparameters for primal averaging and momentum method satisfy (62) and (63), we have the momentum method equivalent to the primal averaging method.

Since the primal averaging method is a special case of `schedule-free` (when $\beta_t \equiv \beta = 1$), the convergence result in Theorem 2.1 gives the convergence for the momentum method whenever $(\alpha_t, \lambda_t)$ in Algorithm 3 satisfies (6), (62), (63). To illustrate this, we start showing the convergence of the momentum method when its stepsize $\{\alpha_t\}_{t=0}^T$ is given by some schedule.

**Corollary D.2.** Let $\{\alpha_t\}_{t=0}^T$ be given by some scheduler. Initializing $\lambda_0, \gamma_0$ such that $(1+\lambda_0)\gamma_0 - \alpha_0 > 0$, consider $\{\gamma_t\}_{t=0}^{T-1}, \{\lambda_t\}_{t=0}^{T-1}$ such that $\gamma_1 = \frac{\alpha_0 \gamma_0}{(1+\lambda_0)\gamma_0 - \alpha_0} > 0$ and for $t = 1, \ldots, T-1$,

$$\lambda_t = \frac{\alpha_{t-1}}{\gamma_t^2}\sum_{i=0}^{t-1}\gamma_i, \quad \gamma_{t+1} = \frac{\alpha_t \sum_{i=0}^t \gamma_i}{\frac{\alpha_{t-1}}{\gamma_t}\left(\sum_{i=0}^{t-1}\gamma_i\right) + \gamma_t - \alpha_t}. \tag{66}$$

Suppose that

$$\alpha_t < \frac{\alpha_{t-1}}{\gamma_t}\left(\sum_{i=0}^{t-1}\gamma_i\right) + \gamma_t. \tag{67}$$

Then Algorithm 3 with parameters $(\alpha_t, \lambda_t)$ for $t = 0, 1 \ldots, T-1$ would then give the convergence

$$\mathbb{E}\left[f(\boldsymbol{x}_T) - f(\boldsymbol{x}_\star)\right] \leq \frac{\frac{1}{2}\|\boldsymbol{x}_0 - \boldsymbol{x}_\star\|^2 + \gamma_0(f(\boldsymbol{x}_0) - f(\boldsymbol{x}_\star))}{\sum_{t=0}^T \gamma_t} + \sum_{t=0}^T \frac{\frac{1}{2}\gamma_t^2 G^2}{\sum_{t=0}^T \gamma_t}. \tag{68}$$

*Proof.* From (6), we know that

$$\frac{1}{c_t} - 1 = \frac{\sum_{i=0}^t \gamma_i}{\gamma_t} - 1 = \frac{\sum_{i=0}^{t-1} \gamma_i}{\gamma_t}.$$

Hence, putting (6) into (62) and (63), we have that

$$\frac{\gamma_0 \gamma_1}{\gamma_0 + \gamma_1} = \frac{\alpha_0}{1 + \lambda_0}, \tag{69}$$

$$\alpha_{t-1} \frac{\sum_{i=0}^{t-1} \gamma_i}{\gamma_t} \frac{1 + \lambda_t}{\lambda_t} = \frac{\alpha_t}{\gamma_{t+1}} \sum_{i=0}^{t+1} \gamma_i, \tag{70}$$

$$\gamma_t^2 = \frac{\alpha_{t-1}}{\lambda_t} \left( \sum_{i=0}^{t-1} \gamma_i \right). \tag{71}$$

We see that (69) gives

$$(1 + \lambda_0) \gamma_0 \gamma_1 = \alpha_0 \gamma_0 + \alpha_0 \gamma_1,$$

which implies

$$\gamma_1 = \frac{\alpha_0 \gamma_0}{(1 + \lambda_0) \gamma_0 - \alpha_0}.$$

Since $(1 + \lambda_0)\gamma_0 - \alpha_0 > 0$, we have $\gamma_1 > 0$. Consider $t = 1, \ldots, T - 1$. Rearranging (71), we can easily obtain

$$\lambda_t = \frac{\alpha_{t-1}}{\gamma_t^2} \sum_{i=0}^{t-1} \gamma_i. \tag{72}$$

For (70), we see that

$$\alpha_{t-1} \frac{\sum_{i=0}^{t-1} \gamma_i}{\gamma_t} \frac{1 + \lambda_t}{\lambda_t} = \frac{\alpha_t}{\gamma_{t+1}} \sum_{i=0}^{t+1} \gamma_i = \frac{\alpha_t}{\gamma_{t+1}} \left( \sum_{i=0}^{t} \gamma_i \right) + \alpha_t. \tag{73}$$

Since (72) implies

$$\frac{1 + \lambda_t}{\lambda_t} = \frac{1}{\lambda_t} + 1 = \frac{\gamma_t^2}{\alpha_{t-1} \left( \sum_{i=0}^{t-1} \gamma_i \right)} + 1, \tag{74}$$

(73) then gives

$$\gamma_t + \alpha_{t-1} \frac{\sum_{i=0}^{t-1} \gamma_i}{\gamma_t} - \alpha_t = \frac{\alpha_t}{\gamma_{t+1}} \left( \sum_{i=0}^{t} \gamma_i \right),$$

and hence,

$$\gamma_{t+1} = \frac{\alpha_t \left( \sum_{i=0}^{t} \gamma_i \right)}{\alpha_{t-1} \frac{\sum_{i=0}^{t-1} \gamma_i}{\gamma_t} + \gamma_t - \alpha_t}.$$

This is positive when

$$\alpha_t < \alpha_{t-1} \frac{\sum_{i=0}^{t-1} \gamma_i}{\gamma_t} + \gamma_t.$$

$$\square$$

Given the stepsize $\alpha_t$ of the momentum method, the lemma suggests the choice of the momentum parameter $\{\lambda_t\}_{t=0}^{T-1}$ such that the last-iterate convergence theory holds. The stepsize $\{\alpha_t\}_{t=0}^{T}$ then defines a set of parameters $\{\gamma_t\}_{t=0}^{T}$, which determines the convergence rate of momentum as shown in (68).

On the other hand, if we set the stepsize $\{\gamma_t\}_{t=0}^{T}$ of the primal averaging following some schedule, we can have a new set of hyperparameters for the momentum method that guarantees the theoretical convergence.

**Corollary D.3.** Let $\{\gamma_t\}_{t=0}^T$ be given by some scheduler. Initializing $\lambda_0 \geq 0$ and $\alpha_0 = \frac{\gamma_0\gamma_1(1+\lambda_0)}{\gamma_0+\gamma_1}$, consider the iterates generated by the momentum algorithm (Algorithm 3) with parameters $(\alpha_t, \lambda_t)$ given by

$$\lambda_t = \alpha_{t-1}\frac{\sum_{i=0}^{t-1}\gamma_i}{\gamma_t^2}, \quad \alpha_t = \alpha_{t-1}\frac{\gamma_{t+1}}{\gamma_t}\frac{\sum_{i=0}^{t-1}\gamma_i}{\sum_{i=0}^{t+1}\gamma_i} + \frac{\gamma_t\gamma_{t+1}}{\sum_{i=0}^{t+1}\gamma_i}. \tag{75}$$

for $t = 1, \ldots, T$. We then have the convergence

$$\mathbb{E}\left[f(\boldsymbol{x}_T) - f(\boldsymbol{x}_\star)\right] \leq \frac{\frac{1}{2}\|\boldsymbol{x}_0 - \boldsymbol{x}_\star\|^2 + \gamma_0(f(\boldsymbol{x}_0) - f(\boldsymbol{x}_\star))}{\sum_{t=0}^T \gamma_t} + \sum_{t=0}^T \frac{\frac{1}{2}\gamma_t^2 G^2}{\sum_{t=0}^T \gamma_t}. \tag{76}$$

Let $D := \|\boldsymbol{x}_0 - \boldsymbol{x}_\star\|$. In particular for the constant learning rate $\gamma_t \equiv \gamma = \frac{D}{G\sqrt{T}}$ gives the rate

$$\mathbb{E}\left[f(\boldsymbol{x}_T) - f(\boldsymbol{x}_\star)\right] \leq \frac{f(\boldsymbol{x}_0) - f(\boldsymbol{x}_\star)}{T} + \frac{DG}{\sqrt{T}}. \tag{77}$$

*Proof.* Putting (6) into (62) and (63), we have (69), (70) and (71) hold. Simply by rearranging terms, we obtain

$$\alpha_0 = \frac{\gamma_0\gamma_1(1+\lambda_0)}{\gamma_0+\gamma_1}, \tag{78}$$

and for $t = 1, \ldots, T-1$,

$$\lambda_t = \alpha_{t-1}\frac{\sum_{i=0}^{t-1}\gamma_i}{\gamma_t^2}, \quad \alpha_t = \alpha_{t-1}\frac{\gamma_{t+1}}{\gamma_t}\frac{\sum_{i=0}^{t-1}\gamma_i}{\sum_{i=0}^{t+1}\gamma_i}\frac{1+\lambda_t}{\lambda_t}.$$

Applying (74), we can simplify this as

$$\lambda_t = \alpha_{t-1}\frac{\sum_{i=0}^{t-1}\gamma_i}{\gamma_t^2}, \quad \alpha_t = \alpha_{t-1}\frac{\gamma_{t+1}}{\gamma_t}\frac{\sum_{i=0}^{t-1}\gamma_i}{\sum_{i=0}^{t+1}\gamma_i} + \frac{\gamma_t\gamma_{t+1}}{\sum_{i=0}^{t+1}\gamma_i}.$$

$\square$

Similarly, if we have $\{\gamma_t\}_{t=0}^T$ given by some schedule, we can derive the stepsize $\alpha_t$ and the momentum parameter $\lambda_t$ for `momenutm` and obtain the convergence bound. Moreover, if $\gamma_t \equiv \gamma = \frac{D}{G\sqrt{T}}$, we can obtain the optimal convergence $\mathcal{O}(\frac{D}{G\sqrt{T}})$ for `momentum`.

# E  EXPERIMENTS: SUPPLEMENTARY MATERIAL

## E.1  IMAGE CLASSIFICATION

We conduct experiments on multiple vision models trained on `CIFAR10` and `CIFAR100`, covering both small-scale (`ResNet-20`) and larger-scale architectures (`Wide ResNet (16-8)`, `DenseNet`). Full details of the architectures and training configurations are provided in Table 1. All experiments are based on the open-source framework `https://github.com/fabian-sp/step-back`, which we extend to include the `Schedule-free` optimizer and to support Group-Norm normalization layers rather than BatchNorm for the `ResNet` [8] and `DenseNet` [9] architectures. As mentioned in Section 4, this is to avoid the complication of writing custom BatchNorm code to approximate batch statistics of the $\boldsymbol{x}$ sequence of `Schedule-free`.

### E.1.1  PREDICTIVE POWER FOR DEEP LEARNING

We train a small `ResNet-20` model on `CIFAR10` and compute the theoretical bound in Theorem 2.1. The norm of stochastic gradients is used as a proxy for the Lipschitz constant $G$, while the

---

[8] `https://github.com/akamaster/pytorch_resnet_cifar10/blob/master/resnet.py`

[9] `https://github.com/weiaicunzai/pytorch-cifar100/blob/master/models/densenet.py`

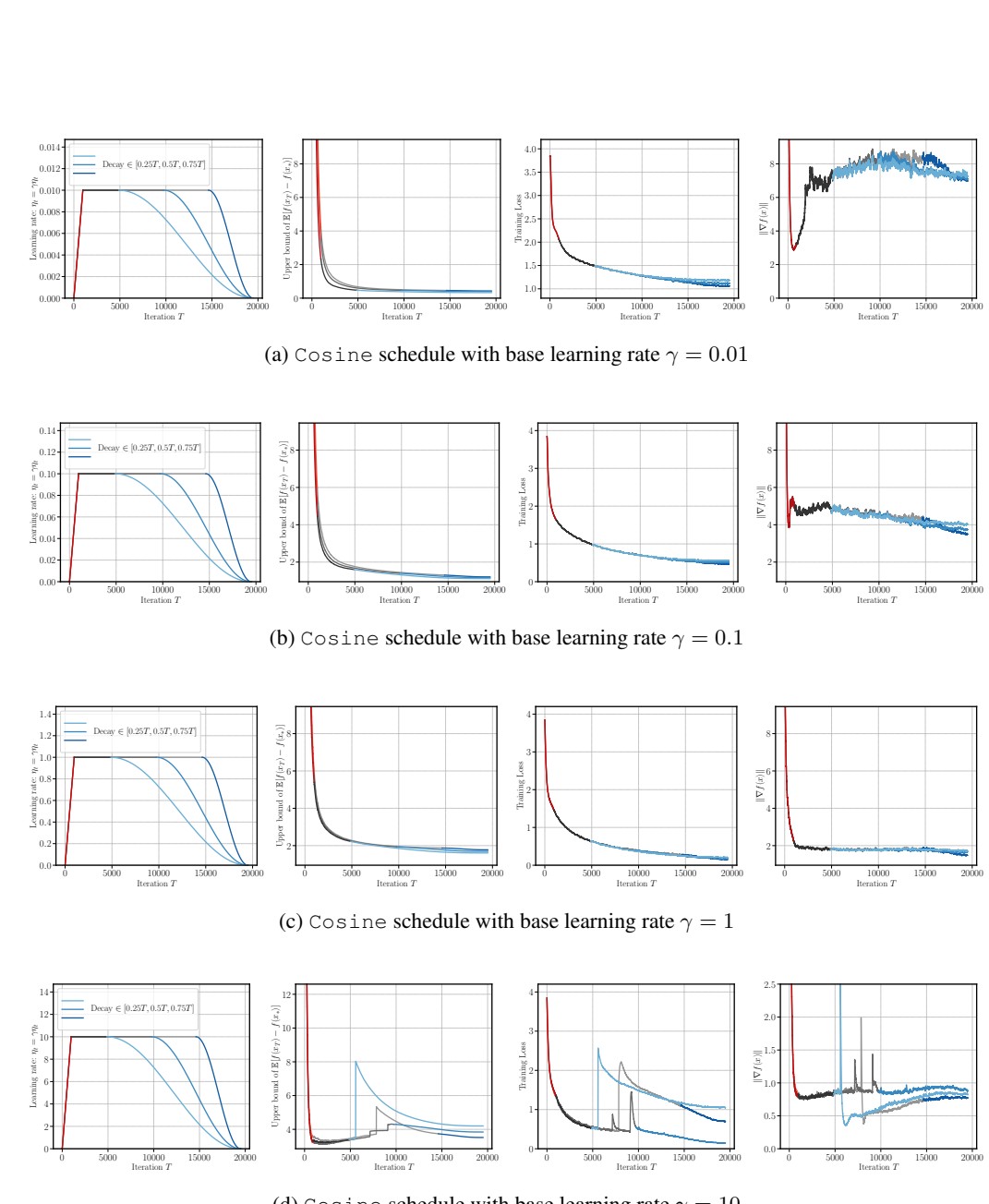

Figure 9: Comparison between the convex theory and the training loss for `cosine` schedule with different cooldown periods and different base learning rates.

| Experiment | `CIFAR10` | `CIFAR100` |
|---|---|---|
| Architectures | `ResNet-20` `Wide ResNet (16-8)` | `DenseNet` |
| Normalization Layer | Group Norm | Group Norm |
| Epochs | 50 | 100 |
| GPUs | $1 \times$ A100 | $1 \times$ A100 |
| Batch size | 128 | 64 |
| Base Learning Rates | [0.01, 0.1, 1, 10] | [0.01, 0.1, 1, 10] |
| Weight Decay | 0.0001 | 0.0002 |
| Momentum | 0.9 | 0.9 |
| Warm-up fraction | 0.05 | 0.05 |
| Cooldown fraction | 0.25 | 0.05 |

Table 1: Comparison of architecture and training setup for image classification on `CIFAR10` and `CIFAR100`.

best parameters and loss during training are used to approximate $x_\star$ and $f(x_\star)$, respectively. We compare our bound for the `wsd` and `cosine` schedules with different cooldown lengths where the decay period begins at iteration $\{0.25T, 0.5T, 0.75T\}$ and $T$ is the training horizon. Figure 3 shows the results for the `wsd` schedule with the base learning rate $\gamma \in \{0.01, 10\}$ and Figure 9 shows the results for the `cosine` schedule with the base learning rate $\gamma \in \{0.01, 0.1, 1, 10\}$. Figure 4 shows the performance of a constant-then-diverging schedule with the base learning rate $\gamma = 10$ and varying diverging lengths.

Since we have discussed Figures 3 and 4 earlier in the paper, we will focus on the discussion over Figure 9 here. In fact, it turns out that both the `wsd` and `cosine` schedules exhibit similar theoretical and empirical performance, so our discussion on the `cosine` schedule is also applicable to `wsd` schedule.

When the base learning rate $\gamma$ is small (i.e., $\gamma \in \{0.01, 0.1, 1\}$), the theory predicts the convergence of the `cosine` schedule well. A slight mismatch is that, earlier cooldown gives a slower empirical convergence, while the theory behaves in the opposite way. We also see that the gradient norm is more stable as $\gamma$ increases. When $\gamma$ is large (i.e., $\gamma = 10$), the theory successfully predicts the spikes in the training loss for different schedules, regardless of whether the spike occurs *before* or *after* the cooldown begins. One possible explanation is that, the spikes in the gradient norms (which is used to approximate the Lipschitz constant $G$ in the theory) lead to the spikes in the theoretical bound. Yet, one should also note that, when $\gamma = 0.01$, the blowup in the gradient norm does not lead to the divergence in the theoretical bound, and both the theoretical bound and the training loss converge.

### E.1.2 STABILITY ANALYSIS

We compare the stability and the performance of `Schedule-free` variants and `SGD-m`. We evaluate both training dynamics and generalization. Models follow the setup in Defazio et al. (2024) for some of the tasks in AlgoPerf: a `Wide ResNet (16-8)` trained on `CIFAR10` (a smaller model) and a `DenseNet` trained on `CIFAR100` (a larger model). Hyperparameters and the setting details are listed in Table 1. We use `wsd` schedule for `SGD-m`, `schedulet` and `Schedule-free` with $c_t = 1/t$ from previous theory, and use `wamrup-stable` schedule only for `Schedule-free` with the heuristic parameters $c_t = \gamma_t^2 / \sum_{i=1}^{t} \gamma_i^2$.

Figure 6 shows the training performance (in terms of the training loss and the validation score) against the learning rate or the number of epochs when training a `Wide ResNet (16-8)` model on the `CIFAR10` data set. We see that, when the learning rate is small, `SGD-m` has a better performance over `schedule-free`, both in terms of the training loss and the validation score. When the learning rate is large, `SGD-m` becomes unstable and `Schedule-free` outperforms `SGD-m`. However, we see that `schedule-free` has a more stable performance in generalization across different learning rates, regardless of the choice of the averaging parameter. In general, `schedulet` has a similar generalization performance as `Schedule-free` with the heuristic averaging parameter $c_t$.

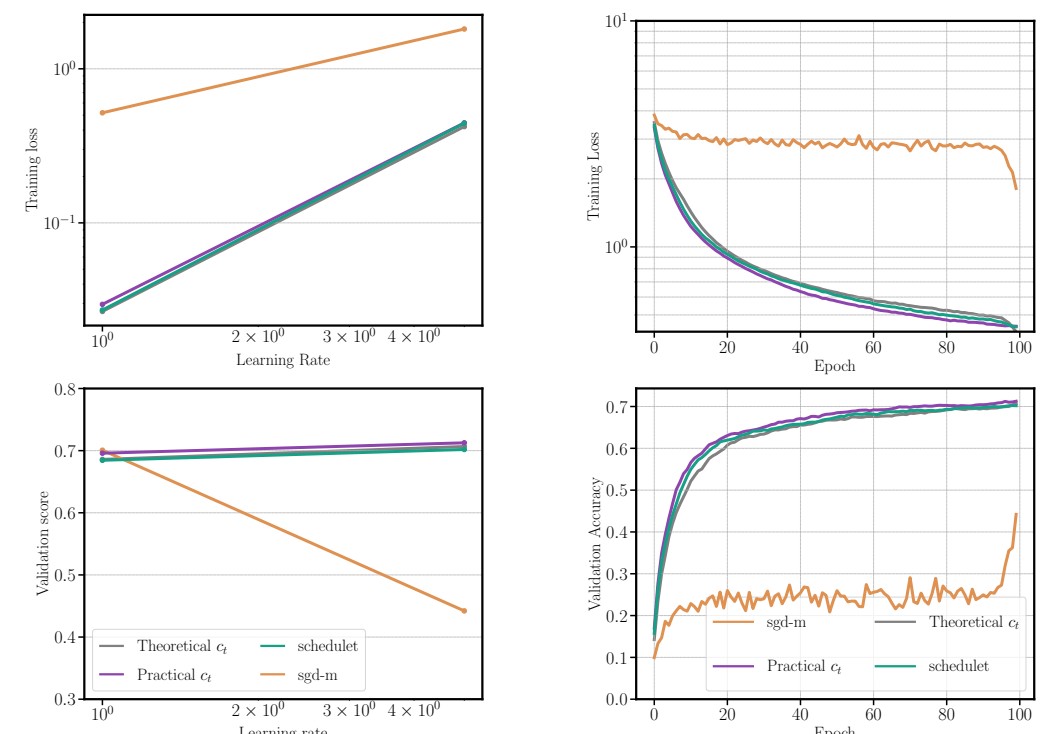

Figure 10: Training a `DenseNet` model on the `CIFAR100` data set.

Figure 10 shows the training performance when training a `DenseNet` model on the `CIFAR100` data set. In this case, `Schedule-free` performs remarkably better than `SGD-m` and is robust over different learning rates. Different choices of averaging parameter $c_t$ have similar performance across different learning rates.

### E.2 BLACK-BOX DISTILLATION DETAILS

Mixed precision training was enabled using `bfloat16` for efficiency. The student model utilized flash attention (Dao et al., 2022).

| Experiment | `tiny_shakespeare` | `fineweb1B` |
|---|---|---|
| Teacher model | `gpt2-medium` | `EleutherAI/gpt-j-6B` |
| Student hidden size | 768 | 768 |
| Student transformer layers | 4 | 12 |
| Student attention heads | 8 | 12 |
| Student vocabulary size | 50257 | 50257 |
| Batch size | 4 | 32 |
| Context length | 512 tokens | 1024 tokens |
| Tokens per training step | 4096 | 262144 |
| Learning rate schedule | Warm-up → Constant → Linear | Warm-up → Constant → Linear |
| Warm-up fraction | 0.1 | 0.1 |
| Cooldown fraction | — | 0.1 |

Table 2: Comparison of model configurations and training setups for distillation on `tiny_shakespeare` and `fineweb1B`.

