# OpenReview forum: "Schedulers for Schedule-free: Theoretically inspired hyperparameters"
_ICLR.cc/2026/Conference — ICLR 2026 Conference Withdrawn Submission_

### Official Review · Reviewer_pAuZ · 2025-10-31

**Soundness:** 3
**Presentation:** 3
**Contribution:** 3
**Rating:** 6
**Confidence:** 3

**Summary:**

This paper extends the theoretical foundation of the schedule-free optimization method by proving last-iterate convergence for arbitrary learning rate schedules (not just constant rates). The key theoretical contribution is showing that the averaging parameter $c_t$ must be set as $c_t = \gamma_t / \sum_{i=0}^t \gamma_i$ to ensure convergence with general schedules. The authors apply this to warmup-stable-decay (wsd) schedules, proving optimal $O(1/\sqrt{T})$ convergence. They also propose schedulep, an adaptive Polyak stepsize variant for schedule-free, with anytime convergence guarantees. Empirically, they demonstrate that their convex theory surprisingly predicts training dynamics on non-convex neural networks and evaluate their methods on image classification and model distillation tasks.

**Strengths:**

**Originality:** The paper makes several novel contributions: (1) extending schedule-free convergence theory beyond constant learning rates to arbitrary schedules, (2) deriving theoretically-motivated averaging parameters $c_t$ as a function of the learning rate schedule, (3) proposing schedulep with Polyak stepsizes for schedule-free achieving anytime optimal convergence, and (4) providing different experiments to compare with real world observations.

**Quality and significance:** The theoretical analysis is rigorous with detailed proofs in the appendix (though I have not verified any proofs). Similarly, I am a fan of work that addresses a genuine gap between theory (e.g. constant schedules) and practice (here warmup schedules for schedule-free methods). Similar to Schaipp et al., the surprising predictive power of convex theory on non-convex problems is noteworthy and could inspire future work.

**Weaknesses:**

There are a few potential concerns about the paper, to me, focused around gap between theory and practice:

* The comparison between theory and practical curves are just qualitative, not quantitative. For instance, for Figure 1, the upper bound is actually increasing before the spikes, whereas the real curves are monotonically decreasing before the spikes.

* (Related to the above and a question below): The theoretical bound needs the parameter G to be simulated. I would therefore be wary to talk of the 'predictive power' of the theory -- the authors themselves use the real runs to establish G, then run the simulation (correct me if I am wrong). I therefore see it more as ad-hoc match rather than 'prediction'.

* The limitations the authors are already aware of: no convergence bounds for the averaging parameter used in practice, and the suggested averaging parameter schedule does not improve the training performance. The other experimental results also seem mixed to me: for instance, adamw-schedulep performs best on shakespeare, but not on fineweb1B; both absolute loss differences do not look strong (see also comment below on the figure clarity).

* More of a conceptual problem for the motivation and real practice: The schedule free method was specifically designed to have _no schedule_. Warmup is purely just used for practical necessity. Reintroducing wsd for schedule free therefore seems of pure theoretical interest to get the optimal convergence rate.

Minor: The authors used the style file of ICLR 2025 :)

**Questions:**

* The figures for LR sweeps are not very clean -- I suggest changing the y-axis scale to better disentangle the differences in losses, or cut some overly bad losses. Potentially increase the fontsize.

* Can you provide quantitative metrics comparing the theory predictions to empirical loss curves, rather than just visual inspection? For example, correlation coefficients or similar?

* To understand correctly: to establish the simulated bounds, you use the real gradient norm statistics to approximate the value of G. This just means you use the upper bound over all observed norms? Or it's done in some online fashion?

* Why restrict to distillation for language model experiments? I would be interested to see even small sized LLM pretraining runs with the different optimizers, and those experiments are not more expensive than distillation. I understand you need a proxy for optimal loss value, but you could equally use a good estimate of the entropy of the text, or the large model's loss value to start with, or ablate how this affects training.

* Connected to the above: How sensitive is schedulep to misspecification of $f_\zeta(x^*)$? Do you have ablations showing potential performance degradation as the approximation quality decreases? And how sensitive is the algorithm with respect to the capping $\gamma_\text{max}$ -- do you have ablations for that?

---

### Official Review · Reviewer_EhPr · 2025-10-31

**Soundness:** 3
**Presentation:** 3
**Contribution:** 2
**Rating:** 2
**Confidence:** 4

**Summary:**

This paper provides some new analysis for optimizer variants using the “schedule-free” style update. First, an analysis of the convergence rate with simple deterministic learning rate schedules is provided, and then an analysis of a more intricate Polyak-inspired learning rate. Some experiments are also included.

**Strengths:**

Polyak theory with optimal constant is interesting.

**Weaknesses:**

I found the Polyak analysis interesting. Although I did not verify the proofs in the appendix in great detail, it seems plausible and achieving the optimal constant is good. However, it suffers from the drawback of all such methods: The interpolation assumption is extremely strong, unless it happens that the interpolation value is zero. Moreover, the result of theorem 3.2 (including the optimal constant) is already achievable without this interpolation assumption by simply setting beta=1 and using a constant learning rate. Of course, beta=1 may not perform as well in practice, but from the perspective of the present theory, beta=1 has the same convergence guarantee while requiring less stringent assumptions.
If the authors were able to provide some more expansive theory on e.g. the hyperparameter robustness of their Polyak method, then this might improve the situation.

Moreover, I found some of the description of Defazio et al to be overly dismissive (which was a bit weird since this paper is heavily based on their work). For example, the phrase “does not guarantee convergence” when applied to Defazio et al Theorem 2 is extremely misleading. This theorem says, essentially, “when a certain quantity is small, then there will be convergence”. It is true that the quantity might not be small, but it is still a convergence guarantee - otherwise none of the results in this paper are convergence guarantees either because the parameters B or G might be large. In fact, many of the results presented here appear to be of the form “here is an algorithm for which the quantity in Defazio et. al Theorem 2 is small”, and so the convergence guarantees presented is actually a corollary of the convergence guarantee of Defazio et al Theorem 2.

This is also another innacurate/misleading characterization of Theorem 2 at the start of section A.5. Contrary to what is written, the setting of c_t = w_t/w_{1:t} is in fact more general than the result presented here because it is *not true* that c_t must have the suggested form of (6).
One could for example set w_t = 1 and gamma_t = 1/sqrt(t), or, more exotically, w_t = t and gamma_t = sqrt(T-t)/t^2. These would all achieve the same O(BG sqrt(T)) convergence result. These only touch on the variations for updating z_t of the same form as discussed in this paper. Defazio et al also suggest more settings for w_t and updates for z_t that achieve e.g. accelerated rates or optimal strongly-convex rates.

The experiments seem relatively small scale and from my reading are not the main focus of this paper.

**Questions:**

Please read and address the weaknesses above. Happy to be corrected if I misunderstand anything.

---

### Official Review · Reviewer_5aSY · 2025-11-01

**Soundness:** 2
**Presentation:** 1
**Contribution:** 2
**Rating:** 2
**Confidence:** 4

**Summary:**

In the paper [1], the authors propose a schedule-free version of SGD where there is an averaging parameter c_t and a learning rate \gamma which is assumed to be constant. They have proved convergence under Lipschitz and convexity assumptions. In Section 2.3, they state that schedule-free does not need a learning rate schedule because it is implicitly applying one.
In this paper the authors apply a polyak learning rate (schedulep) and warmup-stable-decay schedule (schedulet) and derive convergence by obtaining the c_t which will minimize the loss in expectation.

**Strengths:**

see detailed comments under the questions section

**Weaknesses:**

see detailed comments under the questions section

**Questions:**

The paper is not clear in its exposition and I have several questions

 1   The authors use an averaging parameter, c_t = gamma_t/ \sum gamma_t (eq 6) similar to what Defazio uses.
     However, [1] assumes a constant learning rate or a 1/t learning rate, so having this dependency for the averaging parameter c_t over     gamma_t makes sense.  The authors, however, claim that this can be extended to any schedule. Take cyclic schedule or the one-cycle learning rate. The fluctutations in the learning rate are carried over to the averaging parameter; the motivation of averaging the iterates (giving more weightage to recent iterates) is completely upset and the iterates will be all over the place. Can you justify this?

 2  Claim is that this theory (Thm 2.1 specifically) holds for any scheduler. But only results for wsd is shown and cosine annealing (in appendix) . Want to see for more dramatic schedulers, example, super-convergence

3. In Section 5, they explicitly state that their method does not improve the training performance in practice. So I am unsure of the utility of their proposed method apart from predicting the trajectory of training loss in few cases (the figures to the right of Fig. 6 do not seem to agree with theoretically derived values)

4 The paper lacks quantitative evaluation metrics and relies primarily on visual inspection, which the authors themselves acknowledge, thereby limiting confidence in the proposed approach’s robustness.

5    Authors often cite [1]'s use of practical c_t = gamma^2_t/ \sum gamma^2_t. However, they use it only during the warmup period. For the rest, they revert back to 1/t. Is the same thing followed in all of your comparisons?

6    It is not clear what the authors mean by comparing theoretical and empirical convergence. The figures for both are usually on different plots! Does it refer to the fact that the plots converge/do not converge as a binary answer or are they talking about the rate of convergence? Shouldn't they be on the same plot if we want to compare convergence behaviour over training?

7    Line 357 "Theory suggests new choice of c_t" - where?? This is not derived from anywhere. Them 2.1 just says "Suppose that c_t is ----"

General comments:
Figures are very confusing. Legend is presented in an unusual manner.

Fig 1: y axis reads eta_t = gamma eta_t. (???)

Fig 2: there are 2 plots but only one is referenced in the caption

What is the difference between validation score, validation accuracy and final validation loss?

The axes labels are very random

 The ScheduleP method’s reliance on batch statistics, approximated via distillation techniques, poses a limitation when extending it to large language models (LLMs) without distillation.

[1] Defazio, Aaron, et al. "The road less scheduled." Advances in Neural Information Processing Systems 37 (2024): 9974-10007

---

### Official Review · Reviewer_f9UP · 2025-11-04

**Soundness:** 3
**Presentation:** 4
**Contribution:** 3
**Rating:** 6
**Confidence:** 3

**Summary:**

The authors make two primary contributions: first, they extend the last-iterate convergence theory for schedule-free to accommodate arbitrary learning rate schedulers, defining a new averaging parameter update (termed schedulet) that is shown to achieve an optimal $\mathcal{O}(DG/\sqrt{T})$ convergence rate for wsd schedules. Second, they introduce schedulep, a new adaptive Polyak learning rate for schedule-free, and prove its optimal anytime last-iterate convergence. The authors validate these theoretical findings empirically, demonstrating that their convex theory has predictive power for non-convex deep learning tasks and that their novel Polyak schedule performs competitively in black-box model distillation.

**Strengths:**

- The paper rigorously extends the last-iterate convergence theory of schedule-free optimization to accommodate arbitrary, non-constant learning rate schedulers. This is a novel step that closes the significant gap between the method's original theory (limited to constant learning rates) and its practical application, which commonly relies on warmup schedules.
- The paper introduces schedulep, a novel adaptive Polyak learning rate for the schedule-free method. This contribution is practical and well-supported, backed by both an optimal anytime convergence proof and experimental results showing its competitive performance in a black-box model distillation setting.
- Another strength is the empirical demonstration that the new convergence bound, despite being derived for convex functions, possesses predictive power for non-convex deep learning tasks. The authors show it can accurately forecast loss behavior, including transient spikes and divergence.

**Weaknesses:**

1. The new algorithms, schedulet and schedulep, provide limited insights into improving practical performance. Both algorithms are only comparable to existing methods and do not offer significant advantages.
2. The results are restricted to the convex and non-smooth setting. It would be more informative to include results in a broader setting, such as the non-convex and smooth setting. Regarding the predictive power in the convex case, the findings are not particularly surprising. For small step sizes (e.g., in Figure 3), it is intuitively expected that the loss would decrease smoothly. It is not clear from the figure why the prediction follows the actual loss curve more closely than merely an intuition (that for small step size, the loss drops smoothly). For figures exhibiting spikes, it should be noted that the authors used the current stochastic gradient norms as the value of $G$ in the prediction. Consequently, it is natural that the loss exhibits spikes when the gradient norm does. My point is that the predictive power of these spikes does not arise from the theorem itself; other upper bounds with $G$ could predict something similar. The problem is then how to quantitively show the claimed predictive power.
3. The theoretical assumptions should be examined more carefully. In Theorem 2.1, the proof concludes with a term involving $\mathbb{E}\\|g_t\\|^2$. However, in the upper bound, $G^2$ represents the assumed Lipschitz constant of $f$, not of $f_{\zeta}$. It appears that an additional assumption regarding the Lipschitzness of $f_{\zeta}$ is necessary.

Minor issues:
- In some figures, for example, the leftmost subfigure of Figure 1, the label for the $y$-axis is written as $\eta_t = \gamma \eta_t$, which appears to be a typo.
- Figure 5 is presented before Figure 4; it would be clearer to correct the numbering order.
- In Figure 7, the legend in subfigures overlaps considerably with the plotted curves. It is recommended to reposition the legend to improve readability.
- It may be better not to refer to Assumption 3.1 as “interpolation,” since it is considerably weaker than the standard interpolation condition. Using this term could be confusing.

**Questions:**

Please refer to weaknesses 2-3.

In addition:
1. Since the Polyak stepsize requires knowledge of the function value at a minimizer, what happens when only a lower bound of the function value is available instead of the exact one?
2. Theorem 2.1 relies on a specific choice of $c_t$. Are there any negative results showing that other choices of $c_t$, such as the “practical default” or $1/t$, fail to work in theory?

---

### Note · Authors · 2025-11-25

I have read and agree with the venue's withdrawal policy on behalf of myself and my co-authors.